# Helmsman of the Masses? Evaluate the Opinion Leadership of Large Language Models in the Werewolf Game

**Silin Du, Xiaowei Zhang**
Department of Management Science and Engineering
Tsinghua University
Beijing, 100084, China
`{dsl21, xw-zhang21}@mails.tsinghua.edu.cn`

Warning: This work contains examples of potentially unsafe responses from LLMs.

## Abstract

Large language models (LLMs) have exhibited memorable strategic behaviors in social deductive games. However, the significance of *opinion leadership* exhibited by LLM-based agents has been largely overlooked, which is crucial for practical applications in multi-agent and human-AI interaction settings. *Opinion leaders* are individuals who have a noticeable impact on the beliefs and behaviors of others within a social group. In this work, we employ the Werewolf game as a simulation platform to assess the opinion leadership of LLMs. The game includes the role of the Sheriff, tasked with summarizing arguments and recommending decision options, and therefore serves as a credible proxy for an opinion leader. We develop a framework integrating the Sheriff role and devise two novel metrics based on the critical characteristics of opinion leaders. The first metric measures the reliability of the opinion leader, and the second assesses the influence of the opinion leader on other players' decisions. We conduct extensive experiments to evaluate LLMs of different scales. In addition, we collect a Werewolf question-answering dataset (WWQA) to assess and enhance LLM's grasp of the game rules, and we also incorporate human participants for further analysis. The results suggest that the Werewolf game is a suitable test bed to evaluate the opinion leadership of LLMs, and few LLMs possess the capacity for opinion leadership.

⌂ `https://github.com/doslim/Evaluate-the-Opinion-Leadership-of-LLMs`

## 1 Introduction

Recently, large language models (LLMs) have made remarkable improvements and demonstrated a high level of expertise in comprehending and producing human-like natural languages (Jones & Bergen, 2023). However, academically intelligent LLMs are not necessarily socially intelligent (Xu et al., 2024a). The behavioral tendencies of LLMs in social contexts have yet to be explored, which is crucial to coordinate human-AI collaboration and build credible simulations of human society through LLM-based agents (Zhang et al., 2023; Xie et al., 2024; Ziems et al., 2024). Social deductive games, such as Werewolf (Xu et al., 2023; 2024b) and Avalon (Wang et al., 2023a), are suitable scenarios to study the social preference of LLMs (Meng, 2024). Some studies have indicated that LLMs, such as GPT-4 (Achiam et al., 2023), exhibit strategic behaviors including cooperation, confrontation, deception, and persuasion in these games (Xu et al., 2023; 2024b; Lan et al., 2023; Wang et al., 2023a). However, the potential *opinion leadership* of LLMs has been largely overlooked and confounded by existing frameworks and evaluation metrics (Kano et al., 2023). *Opinion leaders* are individuals who exert personal influence on a certain number of other people in certain situations (Rogers & Cartano, 1962). Contrary to the aforementioned interaction strategies, opinion leadership is the composite ability of opinion leaders to comprehensively employ these strategies to influence the decisions of their followers and shape public opinion (Bamakan et al., 2019).

Exploring the degree of opinion leadership in LLM-based agents is crucial for interaction design, decision optimization, public regulation, and the broader field of AI security. In multi-agent systems, such as smart manufacturing, a few opinion leaders can significantly impact task efficiency and outcomes (Rapanos, 2023). As AI assistants and customer service agents, LLMs can shape user experience and business decisions. On social media and forums, the opinion leadership of LLMs could sway social discourse and public opinion. Despite its significance, research on this topic is limited. While there are various methods for identifying opinion leaders within human populations (Li et al., 2013; Bamakan et al., 2019), there is a notable absence of a systematic evaluation of AI agents' opinion leadership, particularly for LLMs.

Examining opinion leadership among LLM-based agents is non-trivial. Opinion leaders are often concealed (Bamakan et al., 2019), making it challenging to identify and distinguish them from other agents and humans in general tasks. Additionally, assessing opinion leadership is also a formidable task due to the complexity of the decision-making process. The impracticality of conducting large-scale real-task randomized controlled trials further complicates the evaluation. Fortunately, the Werewolf game, as a common interactive scenario to study the emergence of AI agent behaviors (Kano et al., 2023), provides a promising testing ground to address these challenges. The *Sheriff* role in the Werewolf game is jointly elected by other players and can summarize the statements and offer decision-making suggestions. This role represents the collective will and allows the player to provide information and exert influence, making it a credible proxy for an opinion leader. By assessing the reliability and spillover of the Sheriffs' decisions, we can evaluate their opinion leadership. Furthermore, incorporating diverse LLMs and human players allows us to compare the impact of LLMs' opinion leadership across different followers. Our main contributions can be summarized as:

1. To the best of our knowledge, this is the first in-depth analysis of opinion leadership within LLMs. We clarify the opinion leadership of diverse LLMs in various contexts.

2. We introduce the setting of the Sheriff and implement a Werewolf game framework, which can seamlessly integrate diverse LLM-based agents and human players. Besides, We devise two novel metrics to evaluate the opinion leadership of different LLMs.

3. We conduct simulations and human evaluations to assess LLMs' opinion leadership, and we collect a Werewolf question-answering (WWQA) dataset for further analysis.

## 2 Related Work

**LLM-based Agent** has risen to prominence in recent research, spurred by advancements in LLMs. These agents are capable of reasoning and decision-making based on their own states (Yao et al., 2023), as well as interacting with the environment (Ahn et al., 2022; Cui et al., 2023). Some studies indicate that LLMs also exhibit a certain level of cognitive abilities (Shapira et al., 2023; Zhuang et al., 2023) and demonstrate potential in simulating believable human behavior (Park et al., 2023). This has led to the development of collaborative frameworks for multi-agent interactions such as AgentVerse (Chen et al., 2023b), AutoGen (Wu et al., 2023), MetaGPT (Hong et al., 2023), ChatEval (Chan et al., 2023), etc. LLM-based agents have shown promise in simulating complex social dynamics in various scenarios, including courtroom (Talebirad & Nadiri, 2023), game development (Hong et al., 2023), auctions (Chen et al., 2023a), etc. To further introduce adversarial dynamics among agents, LLMs are also being utilized in gameplay scenarios such as Texas Hold'em poker (Gupta, 2023), complex video games (Wang et al., 2023b; Zhu et al., 2023), and multi-player social deduction games (such as the Werewolf and Avalon game) (Lan et al., 2023; Light et al., 2023; Wang et al., 2023a; Xu et al., 2023; 2024b; Wu et al., 2024), which heavily rely on natural language interactions. These games provide prototypes of mixed cooperative-competitive social environments similar to human society, allowing researchers to observe whether these agents can exhibit human-like behavior patterns and social principles (Bai et al., 2023; Ren et al., 2024), as well as explore approaches to incentivize the agents to perform better or more human-like. However, existing LLM-based frameworks for the Werewolf game have regrettably overlooked the inclusion of a pivotal character, the "*Sheriff*", resulting in an

underestimation of the influence of opinion leaders. In fact, Avalon also features a similar "leader" role. However, the key difference is that in the Werewolf game, the "Sheriff" is actively elected through agent voting, whereas in Avalon, the leader is passively assigned (which may not fully reflect collective consensus).

**Opinion leadership** is an important concept in the process of social learning in human society (Festinger, 1954). It was first developed in (Lazarsfeld et al., 1944)'s study of the 1940 presidential election. When faced with uncertainty, humans tend to learn from "opinion leaders" to enhance the wisdom of their decisions (Bala & Goyal, 1998). Opinion leaders can be individuals with extensive knowledge in a specific subject area (experts) or individuals with extensive social connections (social connectors) (Goldenberg et al., 2006). They come from the collective will and differ from their followers in information sources, social participation, social status, etc. (Rogers & Cartano, 1962). To exert influence, opinion leaders must be trustworthy (Nahapiet & Ghoshal, 1998), and the extent to which their opinions are adopted by recipients relies on their credibility (Dirks & Ferrin, 2001; Mayer et al., 1995). Trust diminishes conflicts, alleviates the necessity for information verification (Currall & Judge, 1995), and enhances people's inclination and ability to attentively and efficiently embrace others' perspectives (Carley, 1991; Mayer et al., 1995). Xie et al. (2024) found that LLM agents generally exhibit trust behaviors, referred to as agent trust, suggesting the possibility of simulating human trust behaviors with LLM agents. Considering the necessity of trust in opinion leadership, when we strive to create credible simulations of human society, it seems intuitive to extend the concept of opinion leaders to the interactions among LLMs. The agent who garners the most trust and is selected by the majority of agents naturally emerges as the opinion leader. This analysis delves deeper into the social ties among multiple agents, transcending the narrower confines of discussions solely focused on individual agents' strategic behaviors.

## 3 Framework and Proposed Metrics

We consider the Werewolf game with $N = 7$ players, including two Werewolves, three Villagers, one Seer, and one Guard. There is a moderator to move the game along, who will only pass on information and not participate in the game. At the beginning of the game, each player is assigned a hidden role which divides them into the Werewolves and the Villagers (Seer, Guard, and Villager). Then the game alternates between the night round and the day round until one side wins the game. Let $X_i^r$ be the $i$-th player. We use the superscript $r \in \mathcal{R}$ to indicate the role $r in \mathcal{R}$, where $\mathcal{R} = \{W, V, Se, G\}$ represents the Werewolf, Villager, Seer, and Guard, respectively. We might omit the index $i$ or the role $r$ when there is no ambiguity.

We refer to one night-day cycle as a *round*, indexed by $t$. At night, the Werewolves recognize each other and choose one player to kill. The Seer chooses one player to check the player's hidden role. The Guard protects one player, including themselves, from the Werewolves. The Villagers do nothing. We use $A_{night,t}$ to represent all night actions.

During the day, three phases are performed in order: the announcement phase, the discussion phase, and the voting phase. First, the moderator announces last night's result to all players $D_t$. Only the first day has an election phase for the Sheriff. The Sheriff is a special role that does not conflict with the players' roles. The elected Sheriff has the authority to **determine the statement order** and **provide a summary** to persuade other players to vote in alignment with them. Then, each player takes turns to make a statement in an order specified by the Sheriff. At last, each player votes to eliminate one player or chooses not to vote. The actions of making a statement and voting are defined as follows.

$$
\begin{aligned}
X_i \text{ makes a statement:} \quad & S_{i,t} = A_{state,t}(X_i) \\
X_i \text{ votes to eliminate } X_j: \quad & X_j = A_{vote,t}(X_i)
\end{aligned}
$$

After the voting phase, the moderator announces the result $E_t$, and the game moves to the next night's round. The Werewolves win the game if the number of remaining Werewolves equals the combined number of remaining Seer, Guard, and Villagers. The Seer, Guard, and Villagers win if all Werewolves are eliminated.

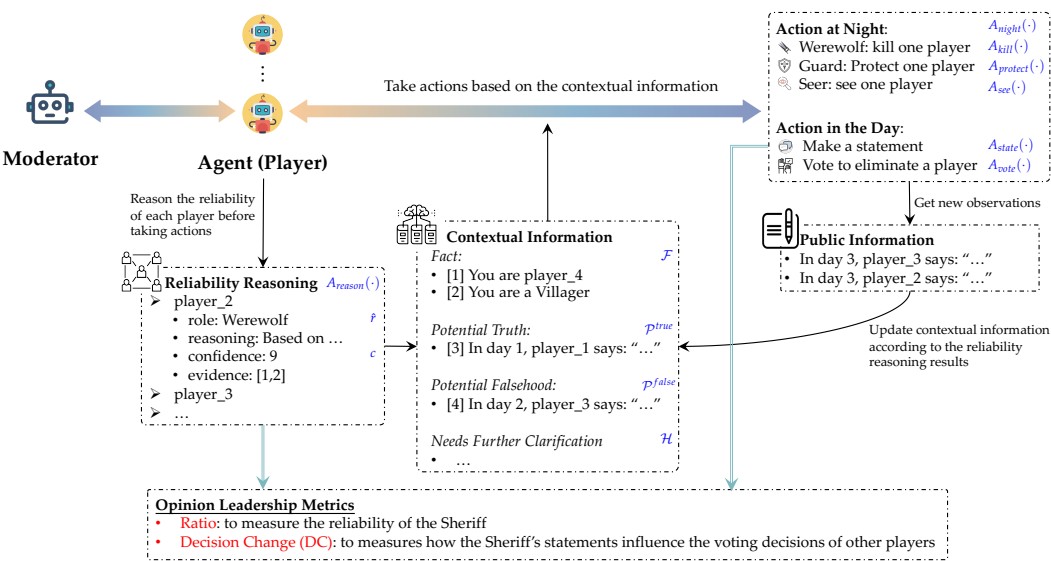

Figure 1: The game framework to evaluate the opinion leadership of LLMs. The blue font shows some simplified notations, with the full list available in Table 16 of Appendix B. Each player needs to reason about the roles and reliability of other players before taking any action. We design two metrics to measure the opinion leadership of the LLM acting as the Sheriff. **Ratio** measures the credibility of the Sheriff, while **DC** assesses the Sheriff's influence on the voting decisions of other players. More details are presented in Section 3.2.

Figure 1 depicts our framework extended from Xu et al. (2024b). The players in our framework can be LLMs or humans. More details about game settings can be found in Appendix A.1 and all notations are listed in Table 16 in Appendix B.

## 3.1 LLM-based Player

Players of different roles are required to perform a series of actions, including night actions (to kill, protect, or see) and day actions (to speak and vote). Each LLM-based player is implemented by an LLM-based agent, which can store the related information during the game and act in response to the game situation.

The LLM-based players perform actions through prompting. The prompt consists of 3 major components: (1) the game rules and the assigned role; (2) the contextual information; and (3) the task description. The prompt templates can be found in Appendix A.2. Communication and action history play a crucial role in the game. However, given the limited context length of LLMs, it might be infeasible to integrate all the information into the prompt. Thus, following Xu et al. (2024b), we design a strategy to manage the contextual information.

To begin with, we divide the contextual information for $X_i^r$ into the following two parts.

1. The set of **facts** $\mathcal{F}_{i,t}$ consists of all factual information before round $t$, including the role of $X_i^r$, the announcement made by the moderator, etc. For example, $\mathcal{F}_{i,t}$ of an Seer $X_i^{\text{Se}}$ can be expressed as

$$
\mathcal{F}_{i,t} = \underbrace{\{i, \text{Se}\}}_{\text{ID \& role}} \cup \underbrace{\{D_\tau\}_{\tau=1}^{t-1}}_{\text{night results}} \cup \underbrace{\{E_\tau\}_{\tau=1}^{t-1}}_{\text{day results}} \cup \underbrace{\{S_{i,\tau}\}_{\tau=1}^{t-1}}_{X_i\text{'s statements}} \cup \underbrace{\{A_{see,\tau}(X_i^{\text{Se}})\}_{\tau=1}^{t-1}}_{X_i\text{'s night actions}}
$$
$$
\bigcup_{k \in \text{Alive}_d(\tau)} \underbrace{\{A_{vote,\tau}(X_k)\}_{\tau=1}^{t-1}}_{\text{voting actions}} \tag{1}
$$

where $\text{Alive}_d(\tau)$ is the IDs of alive players before the day of round $\tau$.

2. The set of **public statements** $\mathcal{P}_{i,t}$ consists of the public statement of other players before the $t$-th round.

$$\mathcal{P}_{i,t} = \bigcup_{k \in \mathtt{Alive}_d(\tau), k \neq i} \left\{ S_{k,\tau} = A_{state,\tau}(X_k) \right\}_{\tau=1}^{t-1} \tag{2}$$

Note that the statements made by other players might be misleading. In light of this, we design a reasoning step to assess the reliability of other players. Based on this reliability, we classify public statements $\mathcal{P}_{i,t}$ as either potential truths $\mathcal{P}_{i,t}^{true}$ or potential falsehoods $\mathcal{P}_{i,t}^{false}$. Concretely, we implement an additional action, called **reliability reasoning**, before making night or day actions. In this step, we prompt the LLM to infer the hidden roles of other players based on historical information and provide confidence levels. For example, at the night of round $t$, $X_i$ reasons the identity $\hat{r}_{i,j,t}^n$ of $X_j$ and provides the confidence level $c_{i,j,t}^n$ before making actions as follows:

$$X_i \text{ reasons the identity of } X_j: \quad c_{i,j,t}^n, \hat{r}_{i,j,t}^n = A_{reason,t}\left(X_i, X_j\right) \tag{3}$$

The superscript $\{n, s, v\}$ denote the reasoning results before the night actions, making a statement and voting, respectively. The confidence level is a scalar from 5 to 10, then the reliability is

$$m_{i,j,t}^n = \begin{cases} 11 - c_{i,j,t}^n, & \text{if } \hat{r}_{i,j,t}^n = W \text{ and } X_i \text{ is not a Werewolf} \\ c_{i,j,t}^n, & \text{otherwise} \end{cases} \tag{4}$$

Given the reliability of other players, $X_i$ will divide the statements in $\mathcal{P}_{i,t}$ into two parts. It regards the statements of players with reliability larger than $\alpha = 6$ as potential truths, otherwise are potential falsehoods.

$$\mathcal{P}_{i,t}^{n,true}, \mathcal{P}_{i,t}^{n,false} = M(\mathcal{P}_{i,t} \mid \alpha, \{m_{i,j,t}^n \mid j \in \mathtt{Alive}_n(t), j \neq i\}) \tag{5}$$

where $M(\cdot \mid \alpha, \{\cdots\})$ is an operator that splits a set of public statements into two disjoint subsets according to the reliability threshold $\alpha$ and $\mathtt{Alive}_n(t)$ is the IDs of alive players before the night of round $t$.

The sequence of events during round $t$ is illustrated in Figure 4 of Appendix B. At the start of round $t$, the set of facts $\mathcal{F}_{i,t}$ and the set of public statements $\mathcal{P}_{i,t}$ are established. $\mathcal{P}_{i,t} = \varnothing$ if $t = 1$ (at the beginning of the game). The LLM-based player will divide the set of public statements $\mathcal{P}_{i,t}$ into $\mathcal{P}_{i,t}^{true}$ and $\mathcal{P}_{i,t}^{false}$ based on the previous reasoning results. Before taking night actions, $X_i$ reasons the reliability of other players.

$$c_{i,j,t}^n, \hat{r}_{i,j,t}^n = A_{reason,t}\left(X_i, X_j \mid \mathcal{F}_{i,t}, \mathcal{P}_{i,t}^{true}, \mathcal{P}_{i,t}^{false}\right), \quad j \in \mathtt{Alive}_n(t), j \neq i \tag{6}$$

Then $X_i$ takes a night action as follows.

$$A_{night,t}\left(X_i \mid \mathcal{F}_{i,t}, \mathcal{P}_{i,t}^{n,true}, \mathcal{P}_{i,t}^{n,false}\right)$$

where $\mathcal{P}_{i,t}^{n,true}, \mathcal{P}_{i,t}^{n,false}$ are sets of potential truth and potential falsehoods obtained as Eq. (5).

On the day of round $t$, the moderator announces $D_t$. Without loss of generality, we assume that $X_i$ was not killed on the night $t$ and the order of statement is the same as the IDs of players. Then $X_i$ will receive a set of public statements made by preceding players, i.e.,

$$\mathcal{H}_{i,t}^s = \{S_{j,t} \mid j \in \mathtt{Alive}_d(t), j < i\}$$

Now the available public statements for $X_i$ become $\mathcal{P}_{i,t}^s = \mathcal{P}_{i,t} \cup \mathcal{H}_{i,t}^s$, where the superscript $s$ signifies the timestamp before $X_i$ makes a statement. Similar to Eq. (6), $X_i$ first performs reliability reasoning and then makes a statement.

$$c_{i,j,t}^s, \hat{r}_{i,j,t}^s = A_{reason,t}\left(X_i, X_j \mid \mathcal{F}_{i,t}, \mathcal{P}_{i,t}^{n,true}, \mathcal{P}_{i,t}^{n,false}, D_t, A_{night,t}(X_i), \mathcal{H}_{i,t}^s\right), j \in \mathtt{Alive}_d(t), j \neq i$$

$$\mathcal{P}_{i,t}^{s,true}, \mathcal{P}_{i,t}^{s,false} = M\left(\mathcal{P}_{i,t}^{s} \mid \alpha, \{m_{i,j,t}^{s} \mid j \in \text{Alive}_d(t), j \neq i\}\right)$$

$$S_{i,t} = A_{state,t}\left(X_i \mid \mathcal{F}_{i,t}, \mathcal{P}_{i,t}^{s,true}, \mathcal{P}_{i,t}^{s,false}, D_t, A_{night,t}(X_i)\right)$$

Subsequently, $X_i$ will collect a set of public statements after its statement, i.e.,

$$\mathcal{H}_{i,t}^{v} = \{S_{j,t} \mid j \in \text{Alive}_d(t), j > i\}$$

When the discussion phase ends, each player begins to vote individually. Now the available public statements for $X_i$ become $\mathcal{P}_{i,t}^{v} = \mathcal{P}_{i,t}^{s} \cup \mathcal{H}_{i,t}^{v}$, where the superscript $v$ indicates the timestamp before $X_i$ votes. Then $X_i$ reassesses the reliability of other players and then votes.

$$c_{i,j,t}^{v}, \hat{r}_{i,j,t}^{v} = A_{reason,t}\left(X_i, X_j \mid \mathcal{F}_{i,t}, \mathcal{P}_{i,t}^{s,true}, \mathcal{P}_{i,t}^{s,false}, D_t, A_{night,t}(X_i), S_{i,t}, \mathcal{H}_{i,t}^{v}\right),$$
$$j \in \text{Alive}_d(t), j \neq i \tag{7}$$

$$\mathcal{P}_{i,t}^{v,true}, \mathcal{P}_{i,t}^{v,false} = M\left(\mathcal{P}_{i,t}^{v} \mid \alpha, \{m_{i,j,t}^{v} \mid j \in \text{Alive}_d(t), j \neq i\}\right) \tag{8}$$

$$A_{vote,t}\left(X_i \mid \mathcal{F}_{i,t}, \mathcal{P}_{i,t}^{v,true}, \mathcal{P}_{i,t}^{v,false}, D_t, A_{night,t}(X_i), S_{i,t}\right) \tag{9}$$

After the voting phase, the moderator discloses the result $E_t$. $X_i$ updates the set of facts by adding their own actions, other players' voting actions, and the night and day results.

$$\mathcal{F}_{i,t+1} = \mathcal{F}_{i,t} \bigcup \{D_t\} \bigcup \{A_{night,t}(X_i)\} \bigcup \{S_{i,t}\} \bigcup_{k \in \text{Alive}_d(t)} \{A_{vote,t}(X_k)\} \bigcup \{E_t\}$$

For the set of public statements, we require the LLM to output evidence when reasoning in round $t$, and any unmentioned statements in $\mathcal{P}_{i,t}^{v,true}, \mathcal{P}_{i,t}^{v,false}$ are removed to get $\widetilde{\mathcal{P}}_{i,t}^{v,true}$ and $\widetilde{\mathcal{P}}_{i,t}^{v,false}$. Finally, we have

$$\mathcal{P}_{i,t+1}^{true} = \widetilde{\mathcal{P}}_{i,1}^{v,true}, \quad \mathcal{P}_{i,t+1}^{false} = \widetilde{\mathcal{P}}_{i,t}^{v,false}, \quad \mathcal{P}_{i,t+1} = \mathcal{P}_{i,t+1}^{true} \bigcup \mathcal{P}_{i,t+1}^{false}$$

The game moves to the next round.

## 3.2 Opinion Leadership Metrics

We introduce the special role, i.e., the Sheriff, into the game framework. On the day of round $t = 1$, the moderator will announce the selection of the Sheriff after the election phase. Let $L(t)$ be the operator that returns the ID of the Sheriff during the day of round $t$. Without loss of generality, we assume that $X_l$ is elected as the Sheriff, i.e., $L(1) = l$. All alive players will receive a special message after the election phase:

> After discussion and a vote, player_$l$ was selected as the Sheriff, who can determine the order of statements, summarize the discussion, and provide advice for voting at last.

In this way, combined with the description in the game rule, we establish the authority of the Sheriff. During the day of round $t$, $X_{L(t)}$ can determine the order of statements. It first performs the reasoning step to obtain $\mathcal{P}_{L(t),t}^{d,true}$ and $\mathcal{P}_{L(t),t}^{d,false}$ similar to Eq. (3)-(5) and then selects its left- or right-hand side player to speak first by

$$\mathcal{O}(t) = A_{order,t}\left(X_{L(t)} \mid \mathcal{F}_{L(t),t}, D_t, \mathcal{P}_{L(t),t}^{d,true}, \mathcal{P}_{L(t),t}^{d,false}\right)$$

The statement order $\mathcal{O}(t)$ will slightly influence the collection of public statements, i.e., $\mathcal{H}^s$ and $\mathcal{H}^v$. But the whole process of making statements and voting remains the same.

In different scenarios and research contexts, various definitions of opinion leaders have been proposed (Chowdhry & Newcomb, 1952; Katz, 2015; Lazarsfeld et al., 1968; Rogers & Cartano, 1962). Despite these diverse definitions, the common characteristics of opinion leaders are mainly reflected in two aspects: (1) they are generally more trustworthy; (2) they

influence the views and even decisions of others. Based on these, we design two evaluation metrics to measure the opinion leadership of LLM-based players.

**Ratio** is to measure the reliability of the Sheriff. When all players finish their statements, the voting phase commences. We collect all the reasoning results before each player's voting action, i.e., $m_{i,j,t}^v$. Then we calculate the average mutual reliability among all players, excluding the Sheriff.

$$\bar{m}_1(t) = \frac{1}{(N_d(t)-1)(N_d(t)-2)} \sum_{i \in \text{Alive}_d(t), i \neq L(t)} \sum_{j \in \text{Alive}_d(t), j \neq L(t), j \neq i} m_{i,j,t}^v$$

where $N_d(t) = |\text{Alive}_d(t)|$ is the number of alive players on the day of round $t$. Similarly, the average reliability of other players towards the Sheriff is

$$\bar{m}_2(t) = \frac{1}{N_d(t)-1} \sum_{i \in \text{Alive}_d(t), i \neq L(t)} m_{i,L(t),t}^v$$

Then the Ratio is defined as:

$$\text{Ratio} = \frac{1}{T} \sum_{t=1}^{T} \frac{\bar{m}_2(t)}{\bar{m}_1(t)}$$

where $T$ is the total number of rounds. A Ratio greater than 1 indicates that the Sheriff's trustworthiness is higher than that of other players; the higher the Ratio value, the stronger the opinion leadership.

**Decision Change (DC)** measures how the Sheriff's statements influence the voting decisions of other players. At the end of the discussion on day $t$, all players except the Sheriff have made a statement. Now all players are required to reason the reliability of the Sheriff as Eq. (7) and make a *pseudo-voting decision* as Eq. (8-9). Note that in Eq. (9), the available set of public statements is $\mathcal{P}_{i,t}^v \setminus \{S_{L(t),t}\}$. We denote the pseudo-voting decision without the public statement from the Sheriff as $A'_{vote,t}(X_i)$. Then the Sheriff makes a statement and all players proceed to the voting phase to yield $A_{vote,t}(X_i)$.

Based on this, we can calculate the proportion of players that change their decision to align with the Sheriff.

$$\text{DC} = \frac{1}{T} \sum_{t=1}^{T} \frac{\sum_{i \in \text{Alive}_d(t), i \neq L(t)} \mathbb{I}\left\{\left(A'_{vote,t}(X_i) \neq A_{vote,t}(X_{L(t)})\right) \text{ AND } \left(A_{vote,t}(X_i) = A_{vote,t}(X_{L(t)})\right)\right\}}{N_d(t)-1}$$

where $\mathbb{I}(\cdot)$ is the indicator function. The higher the DC value, the better the Sheriff's ability to influence other players' decisions, indicating stronger opinion leadership.

Our two proposed metrics are motivated by the fundamental traits of opinion leaders. Although the specific calculation of these two metrics is coupled with our game framework, the underlying concepts are universal and can be easily applied to other similar game frameworks or even adapted to other types of social deductive games because reasoning and decision-making are the core steps in such games. For example, the framework in Xu et al. (2024b) is similar to ours and also contains reasoning steps. The early attempts in LLMs for the Werewolf game (Xu et al., 2023) also explore the trust relationships between players. Similar settings can be found in Wu et al. (2024). In addition, reasoning and decision-making are prevalent steps within the framework of the Avalon game (Wang et al., 2023a; Light et al., 2023; Lan et al., 2023), making our proposed metrics well-suited for such scenarios.

### 3.3 Human Player

Our framework allows human players to participate in the game through text input. The process for human players to engage in the game is similar to that of LLM-based players, as they also need to reason about the roles of other players before taking actions. The moderator will review the game state and historical information to assist players in reasoning and decision-making. The interface for human players is depicted in Figure 5 of Appendix B.

# 4 Experiments

## 4.1 WWQA Dataset

Previous studies have found that certain LLMs may exhibit subpar performance in the context of the Werewolf game (Xu et al., 2023; 2024b; Wu et al., 2024). This can be attributed, in part, to their limited understanding of the game rules and the inherent challenge of fully unleashing their potential capabilities with simple prompts alone. Additionally, the multitude of versions and rules of different Werewolf games can pose a challenge, as LLMs tend to heavily rely on prior knowledge of specific game rules, leading to strategic behaviors that may not align with the given context or even violate the rules.

To address these limitations, we propose a self-instructional framework (Wang et al., 2022) to generate a dedicated Werewolf Question-Answering (WWQA) dataset that directly caters to our game tasks. This WWQA dataset serves as a valuable resource for effectively assessing an LLM's grasp of the fundamental game rules. Furthermore, we utilize this dataset for supervised fine-tuning, enhancing their foundational abilities within our game setting.

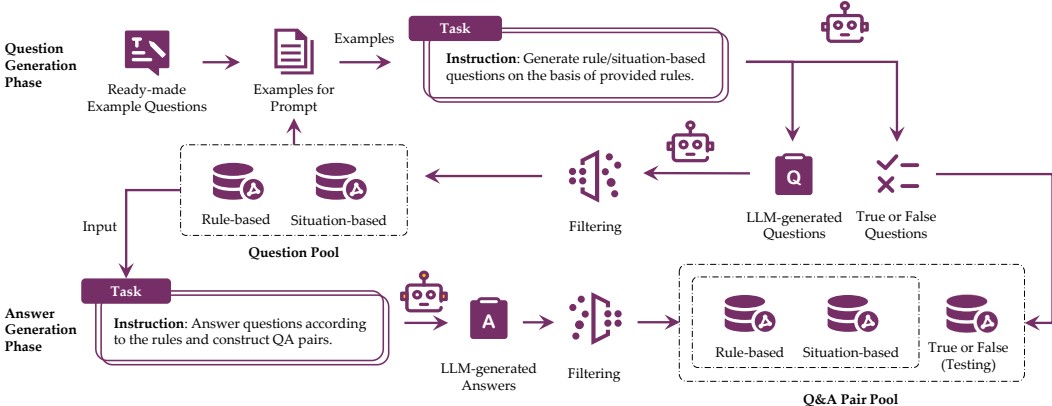

Figure 2: Overview of the data generation process

Figure 2 provides an overview of our data generation process, which includes two primary phases: the question generation phase and the answer generation phase. Utilizing this generation framework, we construct the WWQA dataset. WWQA contains rule-based and situation-based question-answer pairs. Rule-based questions directly target the game rules, while situation-based questions construct a virtual game state, requiring simple reasoning to derive answers. In addition, we collect a *binary QA* dataset for rapid evaluation, where the standard responses are 'Yes' or 'No'. In Appendix C, we offer a detailed description of the data generation process, along with some examples of QA pairs in the WWQA dataset.

## 4.2 Simulation

### 4.2.1 Settings

We consider the simulation first, where all players are LLM-based. To fairly evaluate and compare the opinion leadership of different LLMs, we select a powerful model that can better understand the rules of the Werewolf game as our baseline model. Specifically, we select GLM-3 (Zeng et al., 2022) as the backbone for all players, excluding the Sheriff, which can generate output in a specified format to ensure progression of the game. We provide more details on handling non-compliant output formats in Appendix B.1.

For the Sheriff, we implement it by LLMs of varying sizes from different organizations, including ChatGLM3-6B (C3-6B) (Zeng et al., 2022), Mistral-7B (M-7B) (MistralAI, 2023), Baichuan2-13B (B-13B) (Yang et al., 2023), InternLM-20B (In-20B) (InternLM, 2023), Yi-34B (Young et al., 2024), GLM-3 (Zeng et al., 2022), GLM-4 (ZhipuAI, 2024), and GPT-4 (Achiam et al., 2023). We simulate 30 Werewolf games for each model, with a maximum of 6 rounds. As the Sheriff and non-Sheriff players are implemented by different LLMs,

it's unpredictable whether the tested LLM would be elected. Thus, we omit the election phase. The moderator will secretly select the Sheriff during the initialization of the game roles. This simplification is to fairly compare the opinion leadership of different models. In Appendix D.1, we discuss other experimental settings that incorporate the election phase and provide corresponding simulation results.

In addition to our proposed metrics, we also report some additional indicators in Appendix D.5, such as the win rate, the ratio of players who change their voting decision (not necessarily in agreement with the Sheriff), etc.

### 4.2.2 Results

The results are shown in Table 1. Except for C3-6B, all models achieve decent performance on the binary QA dataset. Overall, the larger the model, the better the results, suggesting a more accurate understanding of the rules. However, in terms of opinion leadership, a larger scale does not mean better results, and open-source models smaller than 20B generally struggle to produce ideal results, with Ratios all below 1 and smaller DC values. Even with the special role of the Sheriff, open-source LLMs cannot gain greater credibility, and only M-7B can change the decision of one player on average. Due to the limited context length (4096), Yi-34B might fail to generate responses as the game deepens, leading to poor performance. Commercially available large-scale LLMs (larger than 100B)[1] perform comparatively better. GLM-3 gains more trust by utilizing the Sheriff's identity, and GLM-4 achieves an accuracy of 0.846 on the binary QA dataset, with a Ratio value reaching 1.167. However, the ability of LLMs to influence the decisions of other players is still relatively weak. We conclude that only a few large-scale LLMs demonstrate a certain degree of opinion leadership. More details and results are provided in Appendix D.

| Metric \ Model | C3-6B | M-7B | B-13B | In-20B | Yi-34B | GLM-3 | GLM-4 | GPT-4 |
|---|---|---|---|---|---|---|---|---|
| **Binary QA** | | | | | | | | |
| Accuracy | 0.582 | 0.756 | 0.750 | 0.794 | 0.792 | 0.760 | 0.846 | 0.850 |
| F1 | 0.565 | 0.753 | 0.749 | 0.789 | 0.794 | 0.761 | 0.846 | 0.851 |
| **Opinion Leadership** | | | | | | | | |
| Ratio | 0.863 | 0.820 | 0.922 | 0.884 | 0.882 | 1.054 | 1.167 | 1.093 |
| DC | 0.088 | 0.151 | 0.118 | 0.068 | 0.037 | 0.126 | 0.113 | 0.107 |

Table 1: Evaluation results on different LLMs

Intuitively, the opinion leadership of LLMs might be improved by enhancing their understanding of the rules. Thus, we use the LoRA (Hu et al., 2021) method with the rank of 8 to fine-tune open-source LLMs with the rule-based and situation-based data from WWQA for 4 epochs. More details are provided in Appendix D.2.

The evaluation results are shown in Table 2, where the models with (FT) are LLMs after fine-tuning. Fine-tuned LLMs generally achieve better results on the binary QA dataset, except for C3(FT). However, a better grasp of the rules does not necessarily equate to better opinion leadership, which is consistent with the results in Table 1. In-20B enjoys a 7.2% increase in Ratio and a 61.8% growth in DC. B-13B gains an 8.6% improvement in Ratio, but its DC value significantly deteriorates. It's non-trivial to improve LLMs' opinion leadership, especially the ability to influence others' decisions.

Furthermore, we further control the identity of the Sheriff. The results in Figure 3 show that LLMs exhibit varying levels of opinion leadership with different roles. When the Sheriff is a Werewolf, it's difficult to gain the trust of other players because more players are on the Villager team. Therefore, all LLMs have poor opinion leadership when they are the Werewolf. In contrast, LLM has stronger opinion leadership with the roles of Seer and Guard, as they can take action at night and gather more information.

---

[1]Both ZhipuAI and OpenAI have not disclosed the specific parameter amounts of the LLMs behind their APIs. Since the base model behind the GLM series is 130B (Zeng et al., 2022) and GPT-3 is 175B (Brown et al., 2020), it's reasonable to say that GLM-3, GLM-4, and GPT-4 are at least 100B.

| Model Metric | C3-6B | C3(FT) | M-7B | M(FT) | B-13B | B(FT) | In-20B | In(FT) |
|---|---|---|---|---|---|---|---|---|
| **Binary QA** | | | | | | | | |
| Accuracy | 0.582 | 0.564 | 0.756 | 0.766 | 0.750 | 0.768 | 0.794 | 0.850 |
| F1 | 0.565 | 0.563 | 0.753 | 0.754 | 0.760 | 0.769 | 0.789 | 0.841 |
| **Opinion Leadership** | | | | | | | | |
| Ratio | 0.863 | 0.847 | 0.820 | 0.779 | 0.922 | 1.002 | 0.884 | 0.948 |
| DC | 0.088 | 0.047 | 0.151 | 0.034 | 0.118 | 0.076 | 0.068 | 0.110 |

Table 2: Evaluation results on fine-tuned LLMs

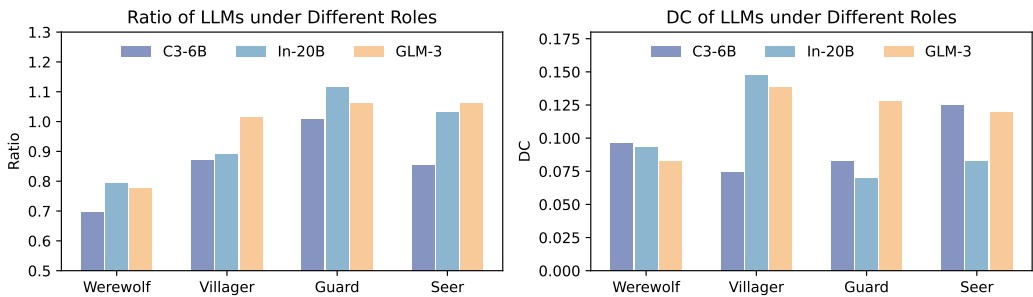

Figure 3: Opinion leadership of LLMs under different roles

## 4.3 Human Evaluation

We invite 8 college students proficient in English and familiar with the Werewolf game for the human evaluation experiment. Each human player will play 5 games with 6 LLM-based players. The process and details can be found in Appendix D.3. We choose GLM-3 as the backbone for LLM-based players, monitor the human player's reasoning and decision-making before and after the Sheriff's statement, and calculate the Spearman correlation coefficient between the human player's reliability scores of the LLM-based player and the scores among other LLM-based players. The results in Table 3 indicate that GLM-3 can gain the trust of human players and demonstrate opinion leadership in human-AI interaction scenarios. Nonetheless, GLM-3 faces challenges in swaying human players with more stringent logic and formidable reasoning skills. Besides, we observe a positive correlation between the reliability scores provided by human players and those inferred by LLM. This partially supports the use of reliability scores provided by LLM-based players during the simulation to compute opinion leadership. Additionally, participants reported hallucination issues with GLM-3 during the experiment, with specific details presented in Appendix D.3.

| Metrics Methods | Ratio | DC | Correlation |
|---|---|---|---|
| Simulation | 1.054 | 0.126 | - |
| Human Evaluation | 1.341 | 0.083 | 0.233 |

Table 3: Human evaluation results

## 5 Conclusion

This study investigates the opinion leadership of LLMs by employing the Werewolf game as a simulation environment. We implement a framework that incorporates a Sheriff role, catering to both LLMs and human players. Besides, we propose two metrics to measure the Sheriff's opinion leadership: Ratio measures their reliability and DC assesses their influence on other players' decisions. Through extensive simulations, we evaluate LLMs of different scales and find that only a few LLMs show a certain degree of opinion leadership. Furthermore, we collect the WWQA dataset to enhance LLMs' grasp of the game rules by fine-tuning. Initial attempts indicate that it's non-trivial to promote the opinion leadership of LLMs. Finally, human evaluation experiments suggest that LLMs can gain human trust, but struggle to influence human decisions.

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

# Appendices

## Contents

# A  Game Rules and Prompt Templates

## A.1  Game Settings

We consider the Werewolf game with $N = 7$ players, including two Werewolves, three Villagers, one Seer, and one Guard. There is a moderator to move the game along, who will only pass on information and not participate in the game. At the beginning of the game, each player is assigned a hidden role which divides them into the Werewolves and the Villagers (Seer, Guard, Villager). Then the game alternates between the night round and the day round until one side wins the game. Let $X_i^r$ be the $i$-th player. We use the superscript $r \in \mathcal{R}$ to indicate the role $r$ in $\mathcal{R}$, where $\mathcal{R} = \{W, V, Se, G\}$ represents the Werewolf, Villager, Seer, and Guard, respectively. For example, $X_3^V$ refers to player_3, who is a Villager. We might omit the index $i$ or the role $r$ when there is no ambiguity.

We refer to a night-day cycle as a *round*, indexed by $t$. At night, the Werewolves recognize each other and choose one player to kill. The Seer chooses one player to check its hidden tole. The Guard protects one player including themselves from the Werewolves. The Villagers do nothing. All night actions in the $t$-th round are formally defined as follows. We use $A_{night,t}$ to represent all night actions.

$$\begin{aligned}
\text{The Werewolf } X^W \text{ kills } X_i: \quad & X_i = A_{kill,t}\left(X^W\right) \\
\text{The Seer } X^S \text{ sees } X_i: \quad & X_i = A_{see,t}\left(X^{Se}\right) \\
\text{The Guard } X^G \text{ protects } X_i: \quad & X_i = A_{protect,t}\left(X^G\right)
\end{aligned}$$

During the day, three phases including an announcement phase, a discussion phase, and a voting phase are performed in order. First, the moderator announces last night's result to all players. We denote the result of the $t$-th night round as $D_t$. Then, each player takes turns to make a statement in an order determined by a special role, the Sheriff. Only the first day has an election phase for the Sheriff. The Sheriff is a special role that does not conflict with the players' roles. The elected Sheriff has the authority to determine the statement order and provide a summary to persuade other players to vote in alignment with him. We use the superscript $*$ to denote the Sheriff, e.g., $X_3^{V,*}$. Then the actions of determining the statement order and making a statement are defined as follows.

$$\begin{aligned}
X^* \text{ determines the statement order:} \quad & [X_i, X_{i+1}, ..., X_N, X_1, ..., X^*] = A_{order,t}\left(X^*\right) \\
X_i \text{ makes a statement:} \quad & S_{i,t} = A_{state,t}\left(X_i\right)
\end{aligned}$$

At last, each player votes to eliminate one player or chooses not to vote, e.g.,

$$X_i \text{ votes to eliminate } X_j: \quad X_j = A_{vote,t}\left(X_i\right)$$

After the voting phase, the moderator announces the result $E_t$ and the game continues to the next night's round. The Werewolves win the game if the number of remaining Werewolves is equal to the number of remaining Seer, Guard, and Villagers. The Seer, Guard, and Villagers win if all Werewolves are eliminated.

The detailed game rules are as follows. We use [Game Rule] to represent them hereafter.

> You are playing a game called Werewolf with some other players. This game is based on text conversations and involves seven players: player_1, player_2, player_3, player_4, player_5, player_6, and player_7. Here are the game rules:
>
> Roles: The moderator, who is also the host, organizes the game, and you must follow their instructions correctly. Do not communicate with the moderator. There are seven roles in the game, including two Werewolves, three Villagers, one Seer, and one Guard. At the beginning of the game, each player is assigned a hidden role, which categorizes them as either a Werewolf or a member of the Village (which includes the Seer, the Guard, and the Villagers). The game then proceeds through alternating night and day rounds until one side emerges victorious.

> During the night round: The Werewolves identify each other and select one player to eliminate; the Seer chooses one player to determine if they are a Werewolf; the Guard selects one player, including themselves, to protect without knowing who the Werewolves have chosen; the Villagers take no action.
>
> During the day round: There are three phases--announcement, discussion, and voting--which are conducted in sequence. In the announcement phase, the results of the previous night are disclosed to all players. If player_i was killed and not protected, the announcement will be "player_i was killed." If a player was killed but protected, the announcement will be "No player was killed."
>
> Only on the first day is there an election phase. In this phase, you can decide whether to nominate yourself for Sheriff based on your role. Players nominated for Sheriff explain their reasons in turn. Each player then votes for one Sheriff nominee or chooses not to vote. The Sheriff is a unique role that does not interfere with a player's primary role. The elected Sheriff has the power to decide which player begins the next discussion round and, as the final speaker, can summarize the discussion to persuade others to vote in agreement with them. In the discussion phase, each remaining player speaks once, in the order determined by the Sheriff, to debate who might be a Werewolf.
>
> In the voting phase, each player votes to eliminate one player or chooses not to vote. The player with the most votes is removed, and the game progresses to the next night's round.
>
> The Werewolves win if their number equals that of the remaining Seer, Guard, and Villagers. The Seer, Guard, and Villagers win by eliminating all the Werewolves.

## A.2  Prompt Templates

We construct our prompt templates following the design in Xu et al. (2024b). The prompt template used to instruct LLMs for reasoning and decision-making consists of three parts.

1. [Game Rule] is illustrated in Appendix A.1.

2. [Contextual Information]. Contextual information includes facts and public statements. Based on the LLM's reasoning results, we categorize public statements into potential truths and potential falsehoods. Table 4 shows an example that organizes contextual information for the model.

3. [Action Instruction] specifies what the LLM needs to accomplish, defining the format of responses to facilitate the parse of the actions. More detailed explanations will be provided later.

Therefore, we leverage [Game Rule] + [Contextual Information] + [Action Instruction] to prompt the LLM to generate responses.

To parse the feedback from LLMs more accurately, we require them to output results in JSON format and provide corresponding output examples. To further reduce the occurrence of invalid actions in LLM outputs (such as selecting a player who has already been eliminated during a night action), we list all available options in the [Action Instruction]. Considering that our experiment involves multilingual LLMs, we explicitly request the language of LLMs' outputs at the end. In addition, we require LLMs to reason about the current situation before taking action.

The template of [Action Instruction] for the Werewolf's night action is shown in Table 5.

The templates of [Action Instruction] for the Guard's and the Seer's night action are shown in Table 6 and 7.

```
All information you can leverage is listed below.
Remaining Players: player_1, player_2, player_4
The following information is true. [𝓕_{i,t} in Eq. (1)]
[1] You are player_1.
[2] You are a Werewolf.
[3] ...

The following information might be true. [𝓟_{i,t}^{n,true}, 𝓟_{i,t}^{s,true}, or 𝓟_{i,t}^{v,true}]
[10] In day 1 round, player_5 said: "Good day, everyone. As the first speaker, I
believe it's crucial to approach this game with a clear head...".
[11] ...

The following information might be false. [𝓟_{i,t}^{n,false}, 𝓟_{i,t}^{s,false}, or 𝓟_{i,t}^{v,false}]
[15] In day 1 round, player_2 said: "Let's carefully consider the information
and behavior of each player during the voting phase to make an informed
decision. Let's work together to eliminate the Werewolves."

The following information still needs further clarification. [𝓗_{i,t}^{s} or 𝓗_{i,t}^{v}]
...
```

Table 4: An example of contextual information

```
Now it is night {} round. As player_{} and a {}, you should choose one player to
kill. You should first reason about the current situation and then choose one of
the following actions: {}.
You should only respond in JSON format as described below.
Response Format:
{
  "reasoning": "reason about the current situation",
  "action": "choose one from {}"
}
Ensure the response is in English and can be parsed by Python json.loads.
```

Table 5: Template of [Action Instrcution] for the Werewolf's night action

```
Now it is night {} round. As player_{} and a {}, you should choose one player to
see. You should first reason about the current situation, then choose one from
the following actions: {}.
You should only respond in JSON format as described below.
Response Format:
{
  "reasoning": "reason about the current situation",
  "action": "choose one from {}"
}
Ensure the response is in English and can be parsed by Python json.loads.
```

Table 6: Template of [Action Instruction] for the Seer's night action

```
Now it is night {} round. As player_{} and a {}, you should choose one player to
protect. You should first reason about the current situation, then choose one
from the following actions: {}.
You should only respond in JSON format as described below.
Response Format:
{
 "reasoning": "reason about the current situation",
 "action": "choose one from {}"
}
Ensure the response is in English and can be parsed by Python json.loads.
```

Table 7: Template of [Action Instruction] for the Guard's night action

```
Now it is day {} discussion phase and it is your turn to speak. As player_{} and
a {}, before speaking to the other players, you should first reason the current
situation only to yourself, and then speak to all other players.
You should only respond in JSON format as described below.
Response Format:
{
 "reasoning": "reason about the current situation",
 "action": "choose one from {}"
}
Ensure the response is in English and can be parsed by Python json.loads. Please
be cautious about revealing your role in this phase.
```

Table 8: Template of [Action Instruction] to make a statement

```
Now it is day {} discussion phase and it is your turn to speak. As player_{} and
a {}, before speaking to the other players, you should first reason the current
situation only to yourself, and then speak to all other players. As the Sheriff,
you can summarize the discussion and provide advice for voting.
You should only respond in JSON format as described below.
Response Format:
{
 "reasoning": "reason about the current situation",
 "action": "choose one from {}"
}
Ensure the response is in English and can be parsed by Python json.loads. Please
be cautious about revealing your role in this phase.
```

Table 9: Template of [Action Instruction] for the Sheriff to make a statement

During the day round, the Werewolf and the Villagers (Villager, Seer, and Guard) share the same template of [Action Instruction] to make a statement, as shown in Table 8. For the Sheriff, the template is slightly different, as shown in Table 9.

In the voting phase, the Werewolf and the Villagers (Villager, Seer, and Guard) leverage different templates of [Action Instruction] to vote, which remind LLMs to maximize the benefit for their team. They are shown in Table 10 and 11, respectively.

```
Now it is day {} voting phase, you should vote to eliminate one player or do not
vote to maximize the Werewolves' benefit. As player_{} and a {}, you should
first reason about the current situation, and then choose from the following
actions: do no vote, {}.
You should only respond in JSON format as described below.
Response Format:
{
 "reasoning": "reason about the current situation",
 "action": "choose one from {}"
}
Ensure the response is in English and can be parsed by Python json.loads.
```

Table 10: Template of [Action Instruction] for Werewolves to vote

```
Now it is day {} voting phase, you should vote to eliminate one player that is
most likely to be a Werewolf or do not vote. As player_{} and a {}, you should
first reason about the current situation, and then choose from the following
actions: do no vote, {}.
You should only respond in JSON format as described below.
Response Format:
{
 "reasoning": "reason about the current situation",
 "action": "choose one from {}"
}
Ensure the response is in English and can be parsed by Python json.loads.
```

Table 11: Template of [Action Instruction] for Villagers to vote

```
Now it is day {} discussion phase and you are the Sheriff. As player_{}, a {}
and the Sheriff, you should first reason the current situation only to yourself,
and then decide on the first player to speak.
You should only respond in JSON format as described below.
Response Format:
{
 "reasoning": "reason about the current situation",
 "action": "choose one from {}"
}
Ensure the response is in English and can be parsed by Python json.loads.
```

Table 12: Template of [Action Instruction] for the Sheriff to determine the order of statements

For the Sheriff, we use the template in Table 12 to determine the order of statements. For simplicity, LLMs only need to decide which player to the left or right speaks first.

All LLM-based players utilize the template in Table 13 to perform reliability reasoning. Due to the fact that some small-scale LLMs cannot follow the predefined JSON template to reason about the reliability of all players at once, the template in Table 13 only requires the LLM to reason about the reliability of one specific player at a time, which enhances the stability of the simulation.

```
Now it is {}, As player_{} and a {}, you should reflect on your previous
deduction and reconsider the hidden roles of {}. You should provide your
reasoning, rate your confidence, and cite all key information as evidence to
support your deduction.  You should only respond in JSON format as described
below.
Response Format:
{
 "{}": {
      "role": select the most likely hidden role of this player from ["Werewolf",
"Seer", "Doctor", "Villager", "Uncertain"],
      "reasoning": your reflection and reasoning,
      "confidence": use an integer from 5 (pure guess) to 10 (absolutely sure)
to rate the confidence of your deduction,
      "evidence": list of integers that cite the key information
 }
}
Ensure the response is in English and can be parsed by Python json.loads.
```

Table 13: Template of [Action Instruction] for reliability reasoning

```
Now it is day 1 election phase, you run for the Sheriff, who can decide the
order of statements, summarize the discussion, and provide advice for voting. As
player_{} and a {}, you should first reason about the current situation, and
then explain why you are qualified to be the Sheriff.
You should only respond in JSON format as described below.
Response Format:
{
 "reasoning": "reason about the current situation only to yourself",
 "statement": "your statement that will be public to all players"
}
Ensure the response is in English and can be parsed by Python json.loads. Please
be cautious about revealing your role in this phase.
```

Table 14: Template of [Action Instruction] for players to make a statement to run for the Sheriff

```
Now it is day 1 election phase, {} are running for the Sheriff, who can decide
the order of statements, summarize the discussion, and provide advice for
voting. As player_{} and a {}, you should first reason about the current
situation, and then vote for a player to be the Sheriff or choose not to vote to
maximize the interests of your team. You should choose from the following
actions: do no vote, {}.
You should only respond in JSON format as described below.
Response Format:
{
 "reasoning": "reason about the current situation",
 "action": "vote for player_i"
}
Ensure the response is in English and can be parsed by Python json.loads.
```

Table 15: Template of [Action Instruction] for players to vote for the Sheriff

Additionally, during the election phase of the first day, we design templates as shown in Tables 14 and 15 to prompt the LLM to make a campaign statement and vote for the Sheriff.

# B   Details of Our Framework

Table 16 summarizes the notations used in our framework. Figure 4 depicts the sequence of events in round $t$. During the night, the Werewolf, Seer, and Guard take turns making decisions. During the day, the moderator announces the results of the previous night $D_t$, and the Sheriff decides the order of speaking. Subsequently, each player makes a statement and votes in turn. Note that the Sheriff always speaks and votes last. Finally, the moderator announces the voting results and the game proceeds to the next round. Each player needs to perform the reliability reasoning step before taking any actions.

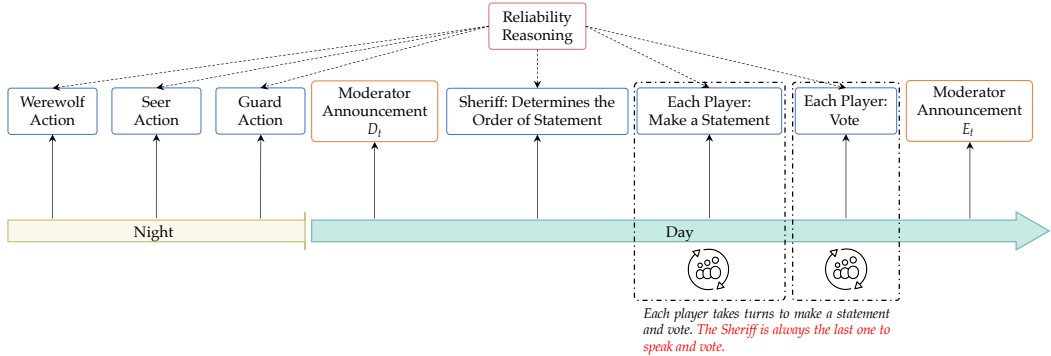

Figure 4: The whole process during round $t$

| Notation | Explanation |
|---|---|
| $N$ | the number of players. |
| $\mathcal{R} = \{W,V,Se,G\}$ | the set of roles denotes the Werewolf, Villager, Seer, and Guard. |
| $X_i^r$ | the $i$-th player with the role $r$ |
| $X^*$ | the Sheriff |
| $t$ | the index of round |
| $A_{kill,t}\left(X^W\right)$ | the night action of the Werewolf |
| $A_{see,t}\left(X^{Se}\right)$ | the night action of the Seer |
| $A_{protect,t}\left(X^G\right)$ | the night action of the Guard |
| $A_{night,t}$ | all night actions |
| $D_t$ | the result of $t$-th night round |
| $A_{order,t}\left(X^*\right)$ | the Sheriff determines the order of statements |
| $A_{state,t}\left(X_i\right)$ | $X_i$ makes a statement |
| $A_{vote,t}\left(X_i\right)$ | the voting action of $X_i$ |
| $A'_{vote,t}\left(X_i\right)$ | the pseudo-voting action of $X_i$ |
| $\mathcal{F}_{i,t}$ | the set of facts of $X_i$ before round $t$ |
| $\texttt{Alive}_d(t)$ | the IDs of alive players before the day of round $t$ |
| $\texttt{Alive}_n(t)$ | the IDs of alive players before the night of round $t$ |
| $N_d(t) = |\texttt{Alive}_d(t)|$ | the number of alive players on the day of round t |
| $\mathcal{P}_{i,t}$ | the available public statement for $X_i$ that contains the statements of other players before the $t$-th round |
| $\mathcal{P}_{i,t}^{true}, \mathcal{P}_{i,t}^{false}$ | the potential truth and potential falsehoods for $X_i$ before the $t$-th round |
| $A_{reason,t}\left(X_i, X_j\right)$ | $X_i$ reasons the role of $X_j$ |
| $\hat{r}_{i,j,t}^n, c_{i,j,t}^n$ | the predicted identity and related confidence level when $X_i$ reasons the role of $X_j$ before taking night action |
| $m_{i,j,t}^n$ | the reliability score of $X_i$ towards $X_j$ before taking night action in round $t$ |
| $\alpha$ | the threshold to divide the potential truths and falsehoods |

| | |
|---|---|
| $M(\cdot \mid \alpha, \{\cdots\})$ | an operator that divides a set of public statements into two disjoint subsets according to the reliability threshold $\alpha$ |
| $\mathcal{H}_{i,t}^{s}$ | the set of public statements made by other players before $X_i$ speaks in round $t$ |
| $\mathcal{P}_{i,t}^{s}$ | the available public statements for $X_i$ before $X_i$ makes a statement in round $t$ |
| $\mathcal{P}_{i,t}^{s,true}, \mathcal{P}_{i,t}^{s,false}$ | the potential truth and potential falsehoods for $X_i$ before makine a statement in the $t$-th round |
| $\hat{r}_{i,j,t}^{s}, c_{i,j,t}^{s}$ | the predicted identity and related confidence level when $X_i$ reasons the role of $X_j$ before making a statement |
| $m_{i,j,t}^{s}$ | the reliability score of $X_i$ towards $X_j$ before making a statement in round $t$ |
| $\mathcal{H}_{i,t}^{v}$ | the set of public statements made by other players AFTER $X_i$ speaks in round $t$ |
| $\mathcal{P}_{i,t}^{v}$ | the available public statements for $X_i$ AFTER $X_i$ makes a statement in round $t$ |
| $\mathcal{P}_{i,t}^{v,true}, \mathcal{P}_{i,t}^{v,false}$ | the potential truth and potential falsehoods for $X_i$ before voting in the $t$-th round |
| $\hat{r}_{i,j,t}^{v}, c_{i,j,t}^{v}$ | the predicted identity and related confidence level when $X_i$ reasons the role of $X_j$ before voting |
| $m_{i,j,t}^{v}$ | the reliability score of $X_i$ towards $X_j$ before voting in round $t$ |
| $\widetilde{\mathcal{P}}_{i,t}^{v,true}, \widetilde{\mathcal{P}}_{i,t}^{v,false}$ | the set of potential truths and potential falsehoods for $X_i$ with unmentioned statements removed at the end of round $t$ |
| $L(t)$ | the operator that returns the ID of the Sheriff during the day of round $t$ |
| $\mathcal{O}(t)$ | the statement order during the day of round $t$ |
| $A_{order,t}\left(X_{L(t)}\right)$ | the Sheriff $X_{L(t)}$ determines the order of statements |
| $\bar{m}_1(t)$ | the average mutual reliability of all players except the Sheriff in round $t$ |
| $\bar{m}_2(t)$ | the average reliability of other players towards the Sheriff in round $t$ |

Table 16: Notations and explanations

## B.1 Resolving Invalid Output

We require LLMs to output results in the specified JSON format, as shown in Appendix A.2, so that we can quickly parse the corresponding content. However, in some cases, LLMs fail to follow the format instructions, and we refer to such outputs as *invalid outputs*. To ensure simulations do not terminate in such situations, we design different mechanisms at various stages of the game to handle invalid outputs. Additionally, if the model cannot generate an output (such as when the prompt exceeds the context window or unsuccessful API calls), or if the model selects an unavailable option, we handle them similarly.

- During the Night. If an LLM-based player who needs to take action at night generates an invalid output, we will randomly select one of all available options as the decision for this player.

- Discussion Phase. If an LLM-based player generates an invalid output when making a statement, we handle it by considering the player to keep silent in the discussion, and other players will receive the public information: "player_{} said nothing".

- Voting Phase. If an LLM-based player generates an invalid output during the voting phase, we consider that the player chooses not to vote.

- Reliability Reasoning. If an LLM-based player produces an invalid output when inferring the identity of $X_i$, the reliability score of $X_i$ will not be updated.

## B.2 Logging System

Our framework includes a comprehensive logging system, which facilitates monitoring the game process during the simulation and checking whether LLM-based players are functioning properly. Specifically, our framework automatically saves three types of logs:

1. Error logs. Error logs primarily store errors that occur during the model generation process, including failures to generate results, outputs that do not meet format requirements, and cases where the model selects an unavailable option.

2. Game logs. Game logs record the progress of the game, including the identities of all players, night actions, the order of speaking during the day, the statements, and voting decisions.

3. Player logs. Player logs record the identities and numbers of players, the contextual information used for decision-making, their statements, decisions (including night actions and voting), and the historical records of their reasoning. Note that we also design a similar player log for human players, which helps us monitor the human evaluation process.

## B.3 Interface During Human Evaluation

Figure 5: Screenshot of the interface during human evaluation

Figure 5 displays the interface during human evaluation. We use different colored fonts to help users distinguish between different types of content. Yellow font is used for prompting user input, red font for announcing night results or voting results, and green font for displaying other players' statements. The input from the human player is displayed in blue on the right side of the screen. Besides, we also use different icons to highlight the source of the messages.

The prompt for requesting input from human players is similar to the one displayed in Appendix A.2, and human players are required to input content in the specified JSON format. Participants are allowed to quickly select the input template using the up and down keys.

## C WWQA Dataset

In this section, we provide a detailed description of the data generation process for the WWQA dataset (Figure 2) and present examples of some QA pairs in the dataset to help readers better understand the structure of our dataset.

### C.1 Data Generation Process

As shown in Figure 2, the data generation process of WWQA includes two primary phases: question generation phase and answer generation phase.

**Question Generation Phase.** In the first phase, our objective is to leverage the generative capabilities of the LLMs to generate a series of game-related questions based on our game rules. Our questions can be categorized into two types based on their content: *rule-based* and *situation-based*. Rule-based questions directly pertain to the game rules. Situation-based questions, on the other hand, require the LLMs to construct a simple virtual game situation, and then engage in a certain level of reasoning within that situation in order to obtain the answer.

We initialize separate example pools, each with a size of 5, for rule-based and situation-based questions. In the first three iterations, we utilize all initial examples along with the game rules to prompt the LLMs to generate 10 questions. The generated questions are then reviewed and corrected in the second round by prompting the LLMs for revisions. The revised questions are then fed into the *question pool*. Starting from the fourth iteration, we sample 2 human-written examples from the *example pool* and 3 LLM-generated questions from the *question pool* separately as examples for prompting the LM in question generation."

**Answer Generation Phase.** In the second phase, our objective is to leverage the combined semantic understanding and generative capabilities of the LLMs to generate concise and logically coherent answers for all the questions in the question pool. Similar to the first phase, we prompt the LLMs for post-processing corrections and filtering to ensure higher-quality answers. Finally, we pair the questions with their corresponding answers to create the final *Q&A Pair Pool*.

In addition, we also prompt the LLMs to generate a *binary QA* dataset, where the standard responses are 'Yes' or 'No'. For this dataset, we conduct additional human review and labeling, using it as a testing set to evaluate LLMs' grasp of our game rules.

### C.2 Examples of QA Pair Data

Table 17 provides some examples of our generated WWQA dataset, which includes three types of question-and-answer data: rule-based, situation-based, and binary QA pairs data.

## D Experiment Details

### D.1 Simulation

We create a set of random seeds for initializing the IDs and roles of each player. For each random seed, we repeat the simulation three times. We consider the following settings.

**1. Heterogeneous Setting:** This is also the setup reported in Section 4.2.

In this setup, we select one LLM as a reference model (e.g., GLM-3) to implement all players except the Sheriff. The Sheriff is implemented using the LLM whose opinion leadership is to be evaluated. It allows for a more fair comparison of the opinion leadership of different LLMs.

To achieve this, we omit the election phase. The moderator will secretly select the Sheriff during the initialization of the game roles and construct the player chosen as the Sheriff

**Examples of Rule-based QA Pairs**

Question: What happens to the player with the most votes in the voting phase?
Answer:   The player with the most votes in the voting phase is eliminated from the game.
Question: How do the Werewolves choose a player to kill?
Answer:   The Werewolves choose a player to kill by voting and selecting the player who receives the most votes.
Question: What actions do the Villagers take during the night round?
Answer:   The Villagers do nothing during the night round.

**Examples of Situation-based QA Pairs**

Question: If the Seer investigates player_2 during the night round and discovers they are a Werewolf, will player_2 be revealed as a Werewolf in the announcement phase?
Answer:   If the Seer investigates player_2 during the night round and discovers they are a Werewolf, player_2 will not be revealed as a Werewolf in the announcement phase. The Seer's findings are not publicly disclosed, and only the actions of the Werewolves (such as killing a player) or the lack of actions (if no one was killed) are announced.
Question: During the discussion phase, if players 1, 3, and 5 suspects player_2, but player_2 is the Seer and reveals their role, how might this affect the voting outcome?
Answer:   During the discussion phase, if players 1, 3, and 5 suspect player_2, but player_2 reveals their role as the Seer, it might affect the voting outcome. The revelation of player_2 being the Seer may lead to a shift in suspicion towards other players, potentially changing the voting decisions.
Question: If player_3 receives the most votes in the voting phase and player_3 is the last remaining Werewolf, will the Werewolves win the game?
Answer:   If player_3 receives the most votes in the voting phase and player_3 is the last remaining Werewolf, the Werewolves will not win the game. The Werewolves win the game if the number of remaining Werewolves is equal to the number of remaining Seer, Guard, and Villagers. In this scenario, if player_3 is the last remaining Werewolf, there would still be Seer, Guard, and Villagers left, and the Werewolves cannot win.

**Examples of Binary QA Pairs**

Question: Is the game won by the Werewolves if all Villagers are eliminated?
Answer:   No
Question: Do the Werewolves know each other?
Answer:   Yes
Question: Can the Seer see the Werewolf who was chosen?
Answer:   No
Question: Is the Sheriff the first speaker in the discussion phase?
Answer:   No
Question: Are there four roles in the Werewolf game?
Answer:   Yes
Question: Can a player who is saved by the Guard be successfully killed by the Werewolves during the night round?
Answer:   No

Table 17: Examples of QA pair data

| Name | Sheriff | Non-Sheriff | Election Phase | Comments |
|------|---------|-------------|----------------|----------|
| **Simulation** | | | | |
| Heterogeneous | Tested LLM | Baseline LLM | $\times$ | |
| Homogeneous | Tested LLM | Tested LLM | $\times$ | |
| Homogeneous Variant 1 | Tested LLM | Tested LLM | $\checkmark$ | |
| Heterogeneous Variant 1 | Tested LLM | Baseline LLM | $\checkmark$ | All players are initialized by the baseline LLM, and the Sheriff is replaced with the tested LLM after the election phase. |
| Heterogeneous Variant 2 | Tested LLM | Baseline LLM | $\checkmark$ | It contains the election phase, and if the tested LLM is not selected as the Sheriff, the simulation ends. |
| **Human Experiment** | | | | |
| Human Evaluation | Tested LLM | Tested LLM & Human Player | $\checkmark$ | |
| Human Baseline | Human Player | Tested LLM | $\times$ | |

Table 18: Summary of different experimental settings

(i.e., $X_l, l = L(1)$) using the LLM to be evaluated. After the first night, the moderator would inform all players of this through the following message.

> After discussion and a vote, player_l was selected as the Sheriff, who can determine the order of statements, summarize the discussion, and provide advice for voting at last.

If the individual chosen covertly as the Sheriff gets eliminated by the Werewolves during the initial night, the current simulation round becomes void. In addition to terminating under the conditions defined in the game rules, a simulation round will immediately stop after the Sheriff is eliminated because only the Sheriff is implemented by the LLM to be evaluated.

**2. Homogeneous Setting:** All players are implemented using the same LLM. The election phase is not implemented in this setting.

**3. Homogeneous Variant 1:** All players are implemented using the same LLM. This setup allows the simulation to proceed strictly according to the game rules.

Due to significant differences in the interaction objects of the Sheriff, it may be problematic to compare opinion leadership metrics under such a setting directly. Nevertheless, we also conduct homogeneous simulations on several LLMs.

On the first day, three players are randomly chosen to participate in the Sheriff's election. After the three players speak, all players vote to decide who will be the Sheriff. When the Sheriff is eliminated, we choose the player the Sheriff trusted the most as the next Sheriff. The game terminates under the conditions defined in the rules. The results after 10 iterations are shown in Table 19.

| Metric \ Model | C3-6B | B-13B | GLM-3 | GLM-4 |
|---|---|---|---|---|
| Ratio | 1.021 | 1.010 | 1.068 | 1.152 |
| DC | 0.122 | 0.108 | 0.118 | 0.098 |

Table 19: Homogeneous Variant 1 results

**4. Heterogeneous Variant 1:** All players are initialized by the same LLM-based agents (default to be GLM-3), and when the election phase is over, the sheriff is replaced with the LLM to be tested.

The results reported in Section 4.2 use a preassigned Sheriff to fairly compare the opinion leadership of different LLMs. In this setting, we consider the fairness of comparison and also introduce the election phase. All players are initialized with the same LLM, and after the election phase, the LLM behind the player elected as Sheriff is replaced with the LLM to be tested. Table 20 shows the relevant results. The results of C3-6B and In-20B are close to those presented in Table 1, indicating that omitting the election process and preassigning the Sheriff has a minimal impact on the evaluation results.

| Metric \ Model | C3-6B (from Table 1) | C3-6B | In-20B (from Table 1) | In-20B |
|---|---|---|---|---|
| Ratio | 0.863 | 0.876 | 0.884 | 0.879 |
| DC | 0.088 | 0.073 | 0.068 | 0.082 |

Table 20: Heterogeneous Variant 1 results

**5. Heterogeneous Variant 2:** One player is implemented by the selected (tested) LLM-based agent while other players are the same LLM-based agents (default to be GLM-3). It contains the election process, and if the LLM to be tested is not elected as the Sheriff, the simulation ends.

In this setting, it is usually difficult to complete a simulation since the tested LLM may fail to be elected as the Sheriff. However, it can be used to estimate the probability of an LLM being elected as the Sheriff under different hidden roles.

### D.2 Fine-tuning

We fine-tune four open-source LLMs on a 3*A100 server using the WWQA dataset. The training set contains 1453 records while the validation set contains 100 records. Specifically, we use the LoRA tuning method Hu et al. (2021) with a rank of 8. The learning rate is $1e-4$ and the batch size is 4. The base models are fine-tuned with causal language modeling tasks over 4 epochs. During the fine-tuning process, the loss on the validation set is shown in Figure 6[2]. We can observe that the loss values of different models decrease and then increase, indicating that models have already overfitted the training data. The loss of LLMs smaller than 10B eventually stabilizes around 0.31, while LLMs larger than 10B can reach a better loss value of about 0.28.

The fine-tuning time for different models varies from 40 to 100 minutes. From the results in Table 2, we conclude that fine-tuning can improve LLMs' understanding of the rules (as evidenced by their performance on the binary QA dataset), but it has a limited effect on enhancing opinion leadership. We need more sophisticated methods to enhance the opinion leadership of LLMs.

---

[2]Since we increase the batch size when fine-tuning Mistral-7B, the global step reduces compared to other models

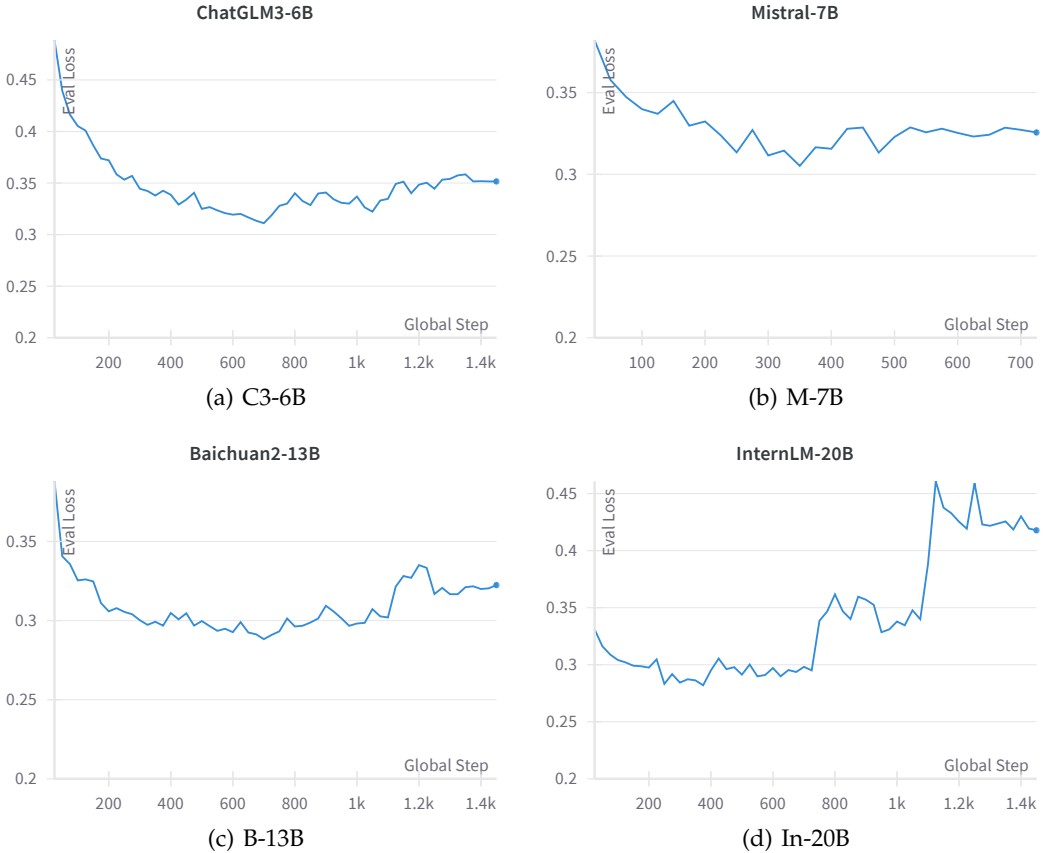

Figure 6: Evaluation loss of different LLMs during fine-tuning

### D.3 Human Evaluation

We invite 8 college students, including 3 graduate students, from a top-tier university in China to participate in the human evaluation. All students have passed the CET-6 English exam and have experience with Werewolf games either online or offline. Our experiment mainly consists of the following three steps.

1. *Training.* We first explain the purpose of the experiment and then confirm with each participant that they understand the settings and rules of the Werewolf game. We extract 5 records from the WWQA dataset to test the participants, ensuring they could participate in the game in English. We also demonstrate how to input content following the format instructions.

2. *Experiment.* Each participant participates in the game by typing text, as shown in Figure 5. Each participant plays 5 iterations with GLM-3. Before each iteration, they should input an integer to initialize the roles of all players. Human players do not participate in the election of the Sheriff and will not be selected as the Sheriff. An iteration of the game ends either according to the rules or immediately if the human player is eliminated.

3. *Interview.* After the experiment, we conduct a simple interview with the participants, asking them to evaluate the LLM's performance in the Werewolf game, including whether the LLM can follow the game rules, whether its logic is rigorous, and whether it's easy to distinguish between LLM-based players and the human players.

The experimental results are shown in Table 3. During the interview, all participants reported that it's easy to recognize LLM-based players since **they always urge other players to provide more information rather than analyzing the game situation**. We display some examples in Appendix E and this can also be found in the game logs of Xu et al. (2023). Besides, human players also mentioned the *hallucination* issues, such as accusing player_6 of remaining silent before player_6's turn to speak. Combined with participants' feedback, we find several examples of hallucination displayed in Table 21.

---

*A hallucination that one can speak during the night.*
player_4: I want to address the concerns raised about my silence last night. As the Seer, I observed player_5 and found out they are a Villager. This information could be useful in identifying the Werewolves. However, I remained silent to avoid providing any information that could potentially put me in danger. I urge everyone to carefully consider the actions and statements of player_6.

---

*A hallucination that some player has already made a statement.*
Moderator: The Sheriff (player_7) chose player_6 to make a statement first in day 1.
player_6: I'm not quite sure who to suspect yet. It's still early in the game and we have limited information. I think player_5's silence is a bit suspicious, but it could just be a strategy.

---

*Time confusion.*
player_3: said nothing.
player_4: player_3 kept silent yesterday, which could be a sign of guilt or fear.

---

*Identity confusion.*
player_7: I encourage everyone to remain cautious and consider all possibilities during the voting phase. player_4's statement in day 2 implicating player_7 and me as Werewolves is concerning.

---

Table 21: Examples of hallucination

## D.4 Human Baseline

Our human evaluation, mentioned in Section 4.3 and Appendix D.3 aims to justify the rationality of the proposed metrics and preliminarily assess the opinion leadership of LLMs in human-AI interactions. There can be multiple ways to conduct further human studies. For example, we extend the framework to set the human player as the Sheriff, which can measure the opinion leadership of humans in such a scenario, serving as a human baseline.

Theoretically, constructing a comprehensive human baseline requires full consideration of the diversity of the human population, such as gender, age, race, education level, etc. Due to time and cost limitations, we only invited a few college students to participate in about 20 rounds of the game. The process is similar to that described in Appendix D.3, and the results are shown in Table 22. Preliminary observations suggest that human players have a stronger ability to influence the decision-making of other players than LLMs. Future research can build a more comprehensive human baseline based on our framework, or conduct more in-depth human experiments.

| Metric | Ratio | DC |
|---|---|---|
| Average Result | 1.231 | 0.226 |

Table 22: Results of the human baseline

## D.5 Additional Metrics

In this subsection, we show several additional metrics derived from the simulations in Section 4.2.

**1. DC$^*$:** A broader metric than DC (Decision Change) that calculates the proportion of players that change their decision after the Sheriff makes a statement. DC$^*$ doesn't require alignment with the Sheriff, which is shown below.

$$\text{DC}^* = \frac{1}{T}\sum_{t=1}^{T} \frac{\sum\limits_{i\in\text{Alive}_d(t), i\neq L(t)} \mathbb{I}\left\{\left(A'_{vote,t}(X_i) \neq A_{vote,t}(X_i)\right)\right\}}{N_d(t) - 1}$$

Table 23 depicts DC$^*$ of different LLMs, and some results are copied from Table 1. The trends of the DC and DC$^*$ before and after fine-tuning are consistent for the same model. Overall, the Sheriff's statements tend to change the decisions of 2-3 players, as they provide additional information. However, only a Sheriff with relatively strong opinion leadership is able to align the choices of other players with their own.

| Metric | C3-6B | C3(FT) | M-7B | M(FT) | B-13B | B(FT) | In-20B | In(FT) | Yi-34B | GLM-3 | GLM-4 | GPT-4 |
|--------|-------|--------|------|-------|-------|-------|--------|--------|--------|-------|-------|-------|
| Ratio | 0.863 | 0.847 | 0.820 | 0.779 | 0.922 | 1.002 | 0.884 | 0.948 | 0.882 | 1.054 | 1.167 | 1.093 |
| DC | 0.088 | 0.047 | 0.151 | 0.034 | 0.118 | 0.076 | 0.068 | 0.110 | 0.037 | 0.126 | 0.113 | 0.107 |
| DC$^*$ | 0.573 | 0.343 | 0.547 | 0.481 | 0.550 | 0.458 | 0.495 | 0.568 | 0.507 | 0.552 | 0.541 | 0.461 |

Table 23: DC$^*$ of different LLMs

**2. Win Rate.** Defining the win rate within our game setting, as described in Section 4.2, is not straightforward. To fairly compare the differences in opinion leadership exhibited by different LLMs, **all players except the Sheriff employ a baseline model (GLM-3)**. If the Sheriff is eliminated, the simulation would automatically end. This is because, in the absence of the sheriff, the remaining models would all be baseline models, rendering it impossible to further evaluate the opinion leadership of the LLM under test. Consequently, these simulations would yield no meaningful game outcomes (win or loss).

The so-called "Win Rate" can also be measured in alternative ways, such as the ratio of games where the Sheriff remains in play and leads the team to victory, and the ratio of games where the Sheriff survives until the end of the game (referred to as the "Completion Rate").

| Metric | C3-6B | C3(FT) | M-7B | M(FT) | B-13B | B(FT) | In-20B | In(FT) | Yi-34B | GLM-3 | GLM-4 | GPT-4 |
|--------|-------|--------|------|-------|-------|-------|--------|--------|--------|-------|-------|-------|
| Ratio | 0.863 | 0.847 | 0.820 | 0.779 | 0.922 | 1.002 | 0.884 | 0.948 | 0.882 | 1.054 | 1.167 | 1.093 |
| DC | 0.088 | 0.047 | 0.151 | 0.034 | 0.118 | 0.076 | 0.068 | 0.110 | 0.037 | 0.126 | 0.113 | 0.107 |
| Completion Rate | 0.267 | 0.200 | 0.200 | 0.300 | 0.433 | 0.467 | 0.567 | 0.533 | 0.500 | 0.400 | 0.433 | 0.400 |
| Win Rate | 0.500 | 0.500 | 0.500 | 0.333 | 0.231 | 0.214 | 0.353 | 0.438 | 0.133 | 0.583 | 0.615 | 0.667 |
| C * W | 0.134 | 0.100 | 0.100 | 0.999 | 0.100 | 0.099 | 0.200 | 0.233 | 0.067 | 0.233 | 0.266 | 0.267 |

Table 24: Win Rate and Completion Rate of different LLMs

From Table 24, we can observe that LLMs with relatively strong opinion leadership (GLM-3, GLM-4, and GPT-4) are more likely to survive until the end of the game and lead their team to victory, while LLMs with weaker opinion leadership (C3-6B and M-7B) may not be able to complete the game. However, this is not strictly correlated.

**3. Ratio of All Roles.** We conduct supplementary experiments under the Homogeneous Setting and report the Ratio for all roles in Table 25. The results show that under different hidden roles, GLM-3 and GLM-4 can generally leverage the Sheriff's identity to gain more trust from other players, especially when playing as the Werewolf.

### D.6 Cost

In addition to the open-source LLMs that can run on our local server, we also extensively utilize the APIs provided by ZhipuAI and OpenAI, with the GLM-3 model being the most frequently called. Our experiment incurs approximately ¥ 1000 for calling the GLM series

| Role
Model | V | V
(Sheriff) | G | G
(Sheriff) | S | S
(Sheriff) | W | W
(Sheriff) |
|---|---|---|---|---|---|---|---|---|
| GLM-3 | 1.010 | 1.023 | 1.014 | 1.093 | 0.992 | 1.045 | 0.903 | 0.996 |
| GLM-4 | 1.043 | 1.015 | 0.969 | 1.067 | 1.007 | 1.051 | 0.958 | 1.054 |

Table 25: Ratio of all roles under the Homogenous Setting

models and $ 25 for calling the GPT-4[3]. A potential cost-saving method is to require the LLM to output the reliability scores of all players at once during the reliability reasoning step. However, this could increase the probability of invalid output, which might affect the simulation progress.

## E   Game Logs

### E.1   Simulation

We display an instance of the game log during the simulation in Table 26, with each LLM-based player implemented by GLM-3.

---

Initialization
```
player_1: Seer
player_2: Guard
player_3: Werewolf
player_4: Villager
player_5: Villager
player_6: Villager
player_7: Werewolf
```

Night 1 Round
```
player_3 chose to kill player_7.
player_7 chose to kill player_1.
player_2 chose to protect player_4.
player_1 chose to see player_3 and player_3 is a Werewolf.
Moderator: No player was killed
```

Comments: *We do not allow Werewolf negotiation due to experimental costs. During small-scale tests, we frequently observed that Werewolves could not reach a consensus after multiple communications. To address this, we introduce sequential decision-making for the Werewolves. One Werewolf makes an initial choice, and then another Werewolf, who knows about this choice, makes their decision accordingly. The difficulty of reaching consensus and our solution are common in similar game settings, e.g. Xu et al. (2023; 2024b).*

Day 1 Discussion Phase
```
Moderator: After discussion and a vote, player_4 was selected as the Sheriff,
who can determine the order of statements, summarize the discussion, and provide
advice for voting at last.
Moderator: The Sheriff (player_4) chose player_5 to make a statement first in
day 1
player_5: I think it's important to consider the role of player_4 as the Sheriff.
They might have Werewolf allies who influenced the vote. We should keep an eye
on them and their actions throughout the game.
player_6: I agree with player_5's concerns about player_4 as the Sheriff. We
should definitely keep an eye on their actions. But for now, I think we should
focus on getting to know each other better and sharing our observations. I will
be watching player_4 closely, and I suggest we all keep our guards up.
```

---

[3]These prices were recorded around February 2024. Due to the continuous changes in the price of LLMs APIs, there may be fluctuations in the cost of experiments.

player_7: I agree with player_6's concerns about player_4 as the Sheriff. We should keep an eye on their actions. However, I think we should also focus on getting to know each other better and sharing our observations. I will be watching player_4 closely, and I suggest we all keep our guards up.
player_1 said nothing.
player_2: I agree with the concerns raised by player_5 and player_6. We should keep a close eye on player_4's actions. But we should also not forget about the other players. player_1 didn't say anything yesterday, which could be a sign of guilt or fear. player_3, player_6, and player_7 all supported player_4, which could mean they are loyal villagers or Werewolves. I will be watching all of them closely.
player_3: I agree with the concerns raised about player_4 as the Sheriff. It's important to keep an eye on their actions. However, I also think we should focus on getting to know each other better. I was surprised that player_1 and player_2 didn't say anything yesterday, which could be a sign of guilt or fear. I will be watching player_4 closely, but I also suggest we all keep our guards up and not forget about the other players.
player_4: I understand the concerns raised about my role as the Sheriff. However, as the elected Sheriff, I assure you that my intentions are pure and I am committed to protecting the Villagers. I agree that it is important to consider the actions of all players, including those who remained silent or supported me. I propose that we continue to observe and discuss each other's behaviors, while also keeping an open mind. Let's work together to uncover the Werewolves' identities and ensure a successful outcome for the Villagers.

---

Comments: *We can observe the hallucination issue in this round of statements.* player_1 *did not speak in this round, but* player_2 *claimed that* player_1 *did not speak 'yesterday'. Meanwhile,* player_3 *mistakenly believed that both* player_1 *and* player_2 *did not speak 'yesterday'.

---

Day 1 Voting Phase
player_5 did not vote.
player_6 voted to eliminate player_2.
player_7 voted to eliminate player_4.
player_1 voted to eliminate player_2.
player_2 did not vote.
player_3 did not vote.
player_4 did not vote.
Moderator: player_2 had the most votes and was eliminated.

---

Night 2 Round
player_3 chose to kill player_4.
player_7 chose to kill player_7.
player_1 chose to see player_4 and player_4 is a Villager.
Moderator: No player was killed

---

Day 2 Discussion Phase
Moderator: The Sheriff (player_4) chose player_5 to make a statement first in day 2
player_5 said nothing.
player_6: I think we should continue to focus on observing player_4's actions as the Sheriff. player_5's silence is concerning, and I suggest we don't forget about player_7's vote for player_4. It's important to keep an open mind and not make hasty judgments. Let's share our observations and work together to uncover the Werewolves' identities.
player_7: Considering the events of the last two days, I believe our focus should be on player_4, given their role as Sheriff and the concerns raised by player_5 and player_6. However, we should not overlook player_5's silence. I suggest we keep an eye on both players and make informed decisions. Let's work together to uncover the Werewolves' identities and ensure a successful outcome for our side.

`player_1` I agree with the concerns raised about player_4 as the Sheriff. However, I was silent yesterday because I wanted to observe and gather more information. I have seen player_3 as a Werewolf and player_4 as a Villager. I suggest we continue to focus on player_4's actions as the Sheriff and keep an eye on player_5 and player_6, who supported player_4. Let's work together to uncover the Werewolves' identities and ensure a successful outcome for the Villagers.
`player_3:` I believe it's important to delve deeper into player_5's silence. While player_7's vote for player_4 is concerning, player_5's lack of input could be a sign of guilt or a strategic move. We should explore why player_5 chose not to speak yesterday. Additionally, I agree with player_6's emphasis on player_4's actions as the Sheriff. However, we should not forget about player_7, who could be playing a quiet but crucial role in the game. Let's continue to observe and discuss, keeping an open mind about everyone's potential roles.
`player_4:` Considering the information we have, it is important to continue observing and discussing the actions of all players. I will be keeping a close eye on player_5 and player_7, as their silence and votes are concerning. However, we should not forget about player_1, player_3, and player_6, as their behaviors could also provide clues about their roles. Let's work together to uncover the Werewolves' identities and ensure a successful outcome for the Villagers.

---

Comments: *In this round of statements,* `player_1`*'s statement revealed himself as a Seer since he directly stated that he had seen the roles of other players.* `player_1` *mentioned that* `player_4` *is a Villager, but he did not strongly support* `player_4`*. Instead, he called on everyone to pay attention to the support from* `player_5` *and* `player_6`*, which is a clear logical flaw. Additionally,* `player_6`*'s statement did not support* `player_4`*, so the statement of* `player_1` *is considered to contain factual errors.*

---

Day 2 Voting Phase
`player_5` did not vote.
`player_6` voted to eliminate player_2.
`player_7` voted to eliminate player_4.
`player_1` voted to eliminate player_2.
`player_2` did not vote.
`player_3` did not vote.
`player_4` did not vote.
`Moderator:` player_2 had the most votes and was eliminated.
The Sheriff was eliminated. This iteration of simulation is over.

---

Comments: *In this round of simulation, the first player to speak (*`player_5`*) questioned the role of* `player_4` *without any evidence, causing other players to develop doubts about* `player_4`*'s identity. This demonstrates that LLMs are easily influenced by the statements of other players when participating in the Werewolf game. Even in the second round, when the Seer (*`player_1`*) revealed* `player_4`*'s true identity, the negative impression of* `player_4` *persisted due to the logical flaws in* `player_1`*'s statement, leading to the simulation ending prematurely.*

Table 26: An example of the game log during the simulation

### E.2 Human Evaluation

We display an instance of the game log during the human evaluation in Table 27, with each LLM-based player implemented by GLM-3.

---

Initialization
`player_1:` Guard
`player_2:` Villager
`player_3:` Villager
`player_4:` Werewolf
`player_5:` Seer

```
player_6: Werewolf
player_7: Villager
Human Player: 4
```

player_4 chose to kill player_5.
player_6 chose to kill player_5.
player_1 chose to protect player_1.
player_5 chose to see player_1 and player_1 is a Guard.
Moderator: player_5 was killed.

Election Phase
Moderator: player_6, player_2, and player_7 are running for the Sheriff. Now they will make a statement in turn.
player_6: I am running for Sheriff because I believe I can bring order to this game and help us Villagers eliminate the Werewolves. I have a strategic mind and can summarize the discussion to guide the voting. Trust me, I will protect our interests as best as I can.
player_2: I am running for Sheriff because I believe I have the ability to bring order to this game and help us Villagers eliminate the Werewolves. I am a keen observer and have taken note of the statements made by the other candidates. I promise to protect our interests and guide the voting process to its most logical conclusion. Trust me, I will do my best to ensure our success.
player_7: I am running for Sheriff because I believe I can bring a fresh perspective to our discussions and help us Villagers make informed decisions. I pay close attention to the details and can summarize the key points to ensure we stay on track. Trust me, I will do my best to protect our interests and guide us toward eliminating the Werewolves.
player_1 voted for player_6.
player_2 voted for player_2.
player_3 did not vote.
player_4 voted for player_6.
player_6 voted for player_2.
player_7 voted for player_2.
Moderator: After discussion and a vote, player_2 was selected as the Sheriff, who can determine the order of statements, summarize the discussion, and provide advice for voting at last.

Comments: *We can observe that* player_6 *concealed their identity during the Sheriff election, pretending to be a member of the Villager team. This is a deceptive strategy that has also been mentioned in Xu et al. (2023); Wang et al. (2023a). However, it is somewhat odd that* player_6 *did not vote for themselves during the voting phase.*

Day 1 Discussion Phase
Moderator: The Sheriff (player_2) chose player_3 to make a statement first in day 1
player_3 said nothing.
player_4: player_3's silence when voting for the Sheriff and during the discussion phase is suspicious, which might be a sign of fear and guilt. I will keep my eye on player_3. Let's share more information and unite together to eliminate the Werewolves.
player_6: player_3's silence during the election phase and discussion phase is quite suspicious. It might be a sign of guilt or fear. Also, player_7's statement seemed a bit too calculated, almost as if they were trying too hard to convince us. I think we should keep an eye on both player_3 and player_7.
player_7: I agree with player_4's concerns about player_3's silence. It is important to be cautious and not accuse anyone without sufficient evidence. I will keep an eye on player_3 and player_6, as their actions and statements seem suspicious. Let's work together to gather more information and eliminate the Werewolves.

player_1: I agree with the concerns raised about player_3's silence. It is important to be cautious and not accuse anyone without sufficient evidence. I will keep an eye on player_3 and player_6, as their actions and statements seem suspicious. Let's work together to gather more information and eliminate the Werewolves.

player_2: Based on the discussion, I think we should pay close attention to player_3, player_6, and player_7. Their actions and statements are suspicious, and we need to gather more information to eliminate the Werewolves. I suggest that we continue to observe their behavior and voting habits. Let's work together to eliminate the Werewolves and bring order to this game.

---

Comments: *We can observe the hallucination issue from this round of statements.* player_6 *accused* player_7*'s speech before* player_7 *made a statement, arousing* player_7*'s suspicion. It led to the suspicion of other players shifting from* player_3 *to* player_6*, a trend also reflected in the subsequent voting phase.*

---

Day 1 Voting Phase

player_3 voted to eliminate player_6.
player_4 voted to eliminate player_3.
player_6 voted to eliminate player_2.
player_7 voted to eliminate player_6.
player_1 voted to eliminate player_6.
player_2 voted to eliminate player_3.
Moderator: player_6 had the most votes and was eliminated.

---

Night 2 Round

player_4 chose to kill player_2.
player_1 chose to protect player_2.
Moderator: No player was killed

---

Day 2 Discussion Phase

Moderator: The Sheriff (player_2) chose player_1 to make a statement first in day 2

player_1: I agree with the concerns raised about player_3's silence. It is important to be cautious and not accuse anyone without sufficient evidence. I will keep an eye on player_3 and player_7, as their actions and statements seem suspicious. Let's work together to gather more information and eliminate the Werewolves.

player_7: I agree with the concerns raised about player_3's silence. It is important to be cautious and not accuse anyone without sufficient evidence. I will keep an eye on player_3 and player_7, as their actions and statements seem suspicious. Let's work together to gather more information and eliminate the Werewolves.

player_4: I agree with player_1 and player_7 that the silence of player_3 is quite suspicious. I protected the Sheriff (player_2) last night since I believe player_2 is a member of our Villager team. Let's keep an eye on player_3 and listen to the analysis of the Sheriff

player_3: I want to thank player_2 for electing me as the Sheriff. I understand the concerns about my silence, but I believe it was a strategic move to observe and gather more information. I agree with the observations about player_7's calculated statements, and I will keep an eye on them. Let's continue to work together, gather more information, and eliminate the Werewolves.

player_2: Based on the discussion, I think we should continue to observe player_7 and player_4, as their actions and statements are suspicious. player_1's agreement with the concerns about player_3's silence makes them a potential ally, but we should remain cautious. Let's work together to gather more information and eliminate the Werewolves.

---

**Comments:** *As the sole Werewolf, the human player (*`player_4`*) chose the Sheriff as the target on the second night but failed. Realizing that the Guard protected* `player_2` *on the second night, the human player disguised themselves as the Guard, claiming to have protected the Sheriff. Taking advantage of their speaking order, they directed suspicion toward* `player_3`*, who had remained silent in the first round.*

**Day 2 Voting Phase**
`player_1` voted to eliminate `player_7`.
`player_7` voted to eliminate `player_4`.
`player_4` voted to eliminate `player_3`.
`player_3` voted to eliminate `player_2`.
`player_2` voted to eliminate `player_3`.
`Moderator`: player_3 had the most votes and was eliminated.

**Night 3 Round**
`player_4` chose to kill `player_2`.
`player_1` chose to protect `player_1`.
`Moderator`: player_2 was killed

**Day 3 Discussion Phase**
`Moderator`: player_2 selected player_1 as the next Sheriff.
`Moderator`: The Sheriff (player_1) chose player_4 to make a statement first in day 3
`player_4`: As the Guard, I protected myself last night. Unfortunately, the previous Sheriff (player_2) was killed last night. In day 2 round, player_2 mentioned that the actions and statements of player_7 were quite suspicious and they selected player_1 as the next Sheriff. Therefore, I believe that player_7 is the active Werewolf.
`player_7`: Given the information we have, I think we should continue to observe player_3 and player_7 closely. Player_4's protection of themselves last night raises questions, and we should keep an eye on their actions. I agree with the concerns raised about player_3's silence, and we should consider their voting patterns carefully. Let's work together to gather more information and eliminate the Werewolves.
`player_1`: Given the information we have, I think it's crucial to continue observing player_3 and player_7, as their actions and statements remain suspicious. Player_4's protection of themselves last night could indicate they are trying to secure their position. We should carefully consider player_3's voting patterns and be cautious of any potential Werewolf influence. Let's work together to gather more information and eliminate the Werewolves.

**Comments:** *The human player (*`player_4`*) continued to disguise themselves as the Guard and used the fact that* `player_4` *had passed the Sheriff to* `player_1` *to redirect suspicion towards* `player_7`*. The remaining two players both displayed hallucinations, mentioning* `player_3`*'s silence, who had already been eliminated in the voting phase of the second day. Furthermore, as the real Guard,* `player_1` *failed to expose* `player_4`*'s disguise.*

**Day 3 Voting Phase**
`player_4` voted to eliminate `player_7`.
`player_7` voted to eliminate `player_7`.
`player_1` voted to eliminate `player_4`.
`Moderator`: player_7 had the most votes and was eliminated.
Game Ends, the Werewolves win.

Table 27: An example of the game log during the human evaluation

