# OpenReview forum: "Helmsman of the Masses? Evaluate the Opinion Leadership of Large Language Models in the Werewolf Game"
_colmweb.org/COLM/2024/Conference — COLM_

### Official Review · Reviewer_H6Wf · 2024-05-11

**Rating:** 6
**Confidence:** 3
**Ethics Flag:** 1

**Summary:**

This paper utilizes the Werewolf game as a simulation platform to gauge the opinion leadership of Large Language Models (LLMs). Within this game, the Sheriff role, responsible for summarizing arguments and suggesting decision options, serves as a credible representation of an opinion leader. The authors construct a framework integrating the Sheriff role and introduce two innovative metrics for evaluation, grounded in the key traits of opinion leaders. The first metric gauges the reliability of the opinion leader, while the second evaluates their influence on other players’ decisions. Through extensive experiments across various scales of LLMs, the authors ascertain their efficacy. Additionally, they construct a Werewolf question-answering dataset (WWQA) to refine LLMs' comprehension of game rules, incorporating human participants for further analysis. Findings suggest that the Werewolf game offers a suitable test bed for assessing the opinion leadership of LLMs, revealing that only a few possess such capacity.

**Questions To Authors:**

1.Why not consider the winning rate of the faction the sheriff belongs to as one of the metrics for measuring Opinion Leadership?

2.To further evaluate the effectiveness of the metrics, why not play a game with only human players, following the same process as a game with only LLM players? In this case, we can use a questionnaire to ask human players their actual thoughts, and then compare them with the Ratio and DC in the game to verify the rationality of the metrics.

3.In the "Night 2 Round" section of Appendix E.1, as far as I know, do the werewolves need to reach a consensus when killing someone? I think it's unreasonable to abandon the kill when opinions differ.

4.Regarding the Ratio metric, it is suggested to present the Ratio values for all roles, not just the sheriff's Ratio. This can help us examine the Opinion Leadership situation of other players. Specifically, for a given simulation, provide the Ratio for all roles.

**Reasons To Accept:**

1.Propose a baseline and design two metrics for measuring Opinion Leadership.

2.Conduct comprehensive experiments, such as human-agent experiments and model fine-tuning.

**Reasons To Reject:**

1.The organization of the paper needs improvement. Section 3 "Framework and Proposed Metrics" is somewhat redundant, and the notation system used in the paper is also overly complex and redundant. This leads to a severe reduction in the content of Section 4. It is recommended to use figures instead of text and enrich Figure 1 to reduce the content of Section 3.

2.Figure 3 in Appendix B is ambiguous, especially the part enclosed by the dashed line. The figure shows that each player votes after making their statement, but in reality, all players should make their statements first, and then vote.

3.See Questions and missing references:

Emergence of Social Norms in Large Language Model-based Agent Societies

Can large language models transform computational social science?

Can Large Language Model Agents Simulate Human Trust Behaviors?

Exploring Collaboration Mechanisms for LLM Agents: A Social Psychology View

---

> ### Author Rebuttal · Authors · 2024-05-31
>
> Thank you for your insightful feedback. We appreciate your comments and will address your concerns as follows:
>
> - W1, W2, \& W3: We are grateful for your practical feedback. We will adjust the organization, modify Figure 3 in Appendix B, and include relevant literature in the final version.
>
> - Q1: We provide a detailed explanation in response to Reviewer 8euA. Win rates are influenced by various factors, and the related calculation in our framework is not straightforward. We have provided related metrics for further discussion in the anonymous link.
>
> - Q2: We appreciate your suggestion and are currently working on it. Due to the significant costs involved in large-scale human evaluations, it is challenging for us to directly engage in extensive human testing. This necessitates a careful examination and justification of comprehensive and reasonable metrics before initiating human evaluations. Building upon our current work, we have started gathering participants on a smaller scale, utilizing methods such as surveys, to conduct a series of human-AI interaction experiments. The aim is to clarify the OL of LLMs through comparisons in various scenarios. We will supplement some preliminary results in our final version.
>
>     Besides, We have also extended the framework to set the human player as the Sheriff for building the human baseline and made results and code available in the link.
>
> - Q3: Thank you for allowing us to clarify this. We do not allow werewolf negotiation due to experimental costs. During small-scale tests, we frequently observed that werewolves could not reach a consensus after multiple communications. To address this, we introduce sequential decision-making for the werewolves. One werewolf makes an initial choice, and then another werewolf, who knows about this choice, makes their decision accordingly. The difficulty of reaching consensus and our solution are common in similar game settings, e.g. arXiv:2309.04658; arXiv:2310.18940 (ICML 2024).
>
> - Q4: We agree with your proposal. However, the OL of the Sheriff and other players is somewhat incomparable since all players except for the Sheriff are based on a baseline LLM. We conduct supplementary experiments under the homogeneous setting and report the ratios for all roles in the anonymous link. Our results show that under different hidden roles, GLM-3 and GLM-4 can generally leverage the Sheriff’s identity to gain more trust from other players, especially when playing as the Werewolf.

---

> > ### Comment · Reviewer_H6Wf · 2024-06-05
> >
> > Thank you for your reply, I have no further questions. I will keep my scores.

---

> > > ### Author Response · Authors · 2024-06-06
> > > **Authors Response**
> > >
> > > Thank you again for your valuable and insightful suggestions! They've helped make our paper better.

---

### Official Review · Reviewer_n5Ra · 2024-05-14

**Rating:** 6
**Confidence:** 3
**Ethics Flag:** 1

**Summary:**

This paper explores the concept of Opinion Leadership in the context of large language models (LLMs) within the social deduction game Werewolf.

The authors develop a framework that integrates the Sheriff role from the game to simulate opinion leadership and propose two novel metrics to evaluate the reliability and influence of the opinion leader. Through simulations and human evaluations, the study assesses the opinion leadership capabilities of various LLMs and finds that only a few demonstrate a certain degree of opinion leadership.

The paper also introduces a Werewolf Question-Answering (WWQA) dataset to enhance LLMs' understanding of the game rules.

**Questions To Authors:**

1. You measure Decision Change (DC) under the setting with and without the Sheriff's statement. Is it possible and meaningful to evaluate the DC with different players being the Sheriff?

2. In this context, would different roles (W, V, Se, G) being the role of the Sheriff have a significant impact on the outcome? Has the evaluation of this question been considered?

3. Typo in Section 3: The Seer chooses one player to check its hidden *tole* --> The Seer chooses one player to check its hidden *role*

**Reasons To Accept:**

1. The paper introduces a novel framework for evaluating opinion leadership in LLMs using the Werewolf game, which is a creative approach to studying AI behaviour in social settings.

2. The authors conduct extensive simulations and include human participants for a more thorough analysis, which strengthens the validity of their findings. The findings have practical implications for the design of AI systems, particularly in multi-agent and human-AI interaction settings.

3. The creation of the WWQA dataset is a valuable contribution to the field, enhancing the models' understanding of game rules and potentially other complex systems.

**Reasons To Reject:**

1. It would be beneficial to gain insights into the influence of Opinion Leadership in this particular or other scenario, such as its impact on Werewolf game outcome orientation. Merely evaluating Opinion Leadership in isolation may seem somewhat meaningless.

2. The opinion leadership metrics may not fully capture the complexity of human social dynamics and opinion formation, which could affect the applicability of the metrics developed.

3. I think there is too much focus on describing the background of this framework, and only a small portion is necessary to help understand the symbols used in the opinion leadership metrics. The remaining information can be moved to an appendix. It might be better to include an introduction to the WWQA dataset and other experimental analyses in the main content.

---

> ### Author Rebuttal · Authors · 2024-05-31
>
> We greatly appreciate your valuable feedback. Regarding your questions, we have provided the following responses. We sincerely hope that our answers adequately address your concerns.
>
> We use "W_" to denote the points mentioned in the "Reason to Reject".
>
> - W1: We fully agree with your point that investigating the impact of opinion leadership (OL) on subsequent performance is important and interesting.
>     However, before delving into deeper exploration, it is imperative that we first validate the existence of OL in LLMs and establish an effective and reliable OL evaluation system, which is the focal point of this paper. Otherwise, any subsequent analysis would be meaningless if LLMs do not exhibit salient OL.
>
>     Besides, OL does not necessarily imply long-term results, similar to many opinion leaders on online forums and social media. Their online influence is immediate and often not outcome-oriented.
>
>     To alleviate your concerns, following your advice, we have added some analyses on whether different LLMs acting as opinion leaders can enhance long-term win rates. Due to space limitations, please refer to further explanations in our reply to Reviewer 8euA.
>
> - W2: We acknowledge the limitations in this aspect. However, capturing and assessing opinion leadership in human society is a highly challenging task. Commonly used methods (such as self-assessment or detecting algorithms) all have their drawbacks in this regard. Nevertheless, our proposed metrics can be extended to other scenarios, such as task-oriented collaboration among multiple LLM agents. Developing a more universal and comprehensive measurement framework will be a focal point of future research. We greatly appreciate your input in this regard.
>
> - Q1 \& Q2: We have examined how the OL of LLMs differ when they play different game roles, as depicted in Figure 2 of the original manuscript. Preliminary evidence suggests that the OL of LLMs is significantly lower when they are the hidden werewolf. We find this comparison intriguing and meaningful, as it mirrors the variations in OL observed in human societies due to differences in social identities.
>
> - W3 \& Q3: We appreciate your detailed suggestions for improving our paper writing. In the final version, we will strive to enhance the description of the framework and provide additional details in other sections.

---

> > ### Comment · Reviewer_n5Ra · 2024-06-04
> >
> > Thanks to the authors for the response. These replies clarify most of my concerns.

---

> > > ### Author Response · Authors · 2024-06-06
> > > **Authors Response**
> > >
> > > Thank you again for your invaluable feedback. Your suggestions are very helpful in improving our paper.

---

### Official Review · Reviewer_8euA · 2024-05-18

**Rating:** 8
**Confidence:** 3
**Ethics Flag:** 1

**Summary:**

The paper evaluates the opinion leadership of large language models (LLMs) using the Werewolf game, a social deduction game.

The study introduces a Sheriff role within the game and introduced two metrics for assessing opinion leadership: reliability and influence on decision-making.

The authors conducted experiments with LLMs of varying scales and incorporated human participants to assess the LLMs' performance in the game.

**Questions To Authors:**

- Incorporating the win rate into the evaluation scheme would also be beneficial, demonstrating how the sheriff's opinion leadership influences the overall voting outcomes, leading the group to either correct or incorrect decisions.


- The background of this framework seems excessive and not essential for understanding the opinion leadership metrics. The introduction of the WWQA dataset could be relocated to the main content for better clarity.

**Reasons To Accept:**

- Well-motivated: Previous studies in such social deduction games mainly focus on the **overall win rate** to show the LLM's ability to deceive or trust. This is the first work (to my knowledge) to **dive into the interations within these games**.
- Well-defined evaluation setup and metric: This paper establishes a well-defined evaluation setup with tailored metrics to measure opinion leadership, incorporating human participants into the framework.

**Reasons To Reject:**

No specific reasons for rejection. A few suggestions are listed below.

---

> ### Author Rebuttal · Authors · 2024-05-31
>
> We appreciate the encouraging feedback and the insightful comments. We have tried to address all the suggestions of the esteemed reviewer.
>
> Q1:
> - The outcome of the game is influenced by numerous factors, such as the actions of werewolves, guards, and seers during the night phase, which are independent of the Sheriff's role. In our framework, the Sheriff primarily contributes to the discussion and voting phase. Therefore, the win rate may not entirely reflect the effectiveness of the Sheriff in online games. In this study, we focus solely on relatively short-sighted measurements. Indeed, as you rightly pointed out, future research could explore whether opinion leadership (OL) contributes to achieving long-term milestones in other scenarios, similar to its manifestation in human society.
>
> - Within the game framework, defining the win rate is not straightforward. We first emphasize a premise: to fairly compare the differences in OL exhibited by different LLMs, **all players except the Sheriff employ a baseline model (GLM-3)**. In our setting, if the Sheriff is eliminated, the simulation will automatically end. This is because, in the absence of the sheriff, the remaining models would all be baseline models, rendering it impossible to further evaluate the OL of the model under test. Consequently, these simulations would yield no actual game outcomes (win or loss).
>
> - The so-called "Win Rate" can also be measured in alternative ways, such as the ratio of games where the Sheriff remains in play and leads the team to victory, and the ratio of games where the Sheriff survives until the end of the game (called the "Completion Rate"). We offer these two metrics in the anonymous link in the paper.
>
> - LLMs with relatively strong OL (GLM-3/GLM-4/GPT-4) are more likely to survive until the end of the game and lead their team to victory, while LLMs with weaker OL (C3-6B/M-7B) may not be able to complete the game. However, this is not strictly correlated.
>
> Q2: We agree with your suggestions for our paper's composition. In the final version, we will try to refine the framework description and add more details to other sections.

---

> > ### Comment · Reviewer_8euA · 2024-06-03
> > **Reviewer Response**
> >
> > Thank you for addressing my earlier questions. Your response has provided me with a deeper understanding of your paper.
> >
> > Based on your clarifications, I am increasing my rating to an 8, which supports clear acceptance.

---

> > > ### Author Response · Authors · 2024-06-06
> > > **Authors Response**
> > >
> > > Thank you again for your endorsement and invaluable advice! Your feedback has given us tremendous confidence and played an essential role in refining our paper.

---

### Official Review · Reviewer_fuFN · 2024-05-22

**Rating:** 6
**Confidence:** 3
**Ethics Flag:** 1

**Summary:**

This paper mainly conducts an analysis on opinion leadership in LLMs under the scenario of Werewolf game. It presents two metrics to evaluate if the LLMs (*Sheriff*) has the capability to lead, or even change the opinion of others. It also proposes a Werewolf game framework, which supports both LLMs and humans to play the game in text. Currently LLMs show limited capability on opinion leadership.

**Questions To Authors:**

Q1. In Table 1, most of the Ratios are either below 1 or closely above 1. If I understand correctly, when the ratio equals 1, it means your words have the average impact on other players' decisions. Does it mean that most LLMs are not either negatively or positively affecting the decisions of other players?

Q2. I feel the two metrics are well defined and make sense, but meanwhile, there are some confusing points. For example, I may miss some details, but what happens if all the players have initially voted for the same player that the Sheriff proposes to vote? There is no decision change, but it seems the Sheriff somewhat consolidates players' minds.

Q3. Opinion leadership may not equal to always persuading others to follow the same trend as the leader. Aggressively questioning one player's role may encourage others to critically evaluate their initial assumptions. Is it possible to consider such cases?

Q4. It is a bit hard to properly feel the scale of DC through the current results. In Table 1, most of the DCs are around 0.1. Does that mean for a game with 6 rounds, the sheriff can only change 0 or 1 player's mind in the whole game?

**Reasons To Accept:**

1. **Writing**: I love the writing. It's pretty well written, reader-friendly and easy to follow. Many details are provided in a clear way.
2. **Framework and dataset contribution**: This paper provides a Werewolf game framework which helps to evaluate LLMs under a multi-turn, interactive complex collaboration and competition scenario.
3. **Objective**: I agree in general that the studies on opinion leadership in LLMs are quite important for AI safety.

**Reasons To Reject:**

W1. As the authors claim in their introduction, opinion leaders should emerge from collective consensus. However, according to the appendix, the Sheriff is preassigned instead of elected by players in the game. A preassigned leader makes less sense to follow or to trust. That may affect the validity of the related result.

W2. There are so many potential angles to go deep in, but it seems the authors just pause after the first step. I would like to see more analyses on logs, e.g. how different it is when the sheriff is a werewolf vs. a seer/guard/villager. Are there any common trends/phenomenons that the discussion may follow (e.g. players may tend to be careful or aggressive / can deceive well or badly, ...), and how these are correlated to the capability of LLMs to lead the opinions?

W3. I think human study in this work is quite important. There can be multiple ways to conduct human studies (following W2). One way is to let humans play as the sheriff, which works as a human baseline that can be directly compared with all the LLM baselines in Table 1. The other possible way is to let one LLM play as the Sheriff and leave the rest of the players controlled by humans. We may or may not let the humans know in advance that Sheriff is controlled by AI. Then we can see humans' inclination to trust the AI or not, which can directly evaluate the LLM's capability of leading opinions in real cases. I feel a bit lost about the reason for designing the human evaluation in the current way.

---

> ### Author Rebuttal · Authors · 2024-05-31
>
> We appreciate your endorsement and valuable suggestions. In response to your concerns:
> - W1: To compare LLMs’ Opinion Leadership (OL) fairly, we use a baseline LLM for non-Sheriff players, making it unpredictable if the tested LLM would be elected Sheriff. Preassigned Sheriffs are used in some experiments, while others, like human evaluation, include an Election Phase (EP). We introduce the EP under heterogeneous settings with two methods: (1) Initialize all players with the baseline LLM and switch the elected Sheriff’s backend to the tested LLM after the EP; (2) Restart if the tested LLM isn’t elected. We’ve shared the code and preliminary results of the first method via the anonymous link in the paper.
> - W2: We strongly agree with you. We believe that examining the OL of LLMs entails a series of efforts and is non-trivial. This paper is an early attempt, aiming to: 1) initiate discussions on this issue; 2) establish a scenario-specific framework; and 3) propose metrics to evaluate OL. We will open-source more game logs and provide further analyses in the next version.
> - W3: Thank you for your suggestions. Our human evaluation justifies the proposed metrics and assesses the OL of LLMs in human-AI interactions. We’ve extended the framework for human players as Sheriffs in small-scale experiments. The results and code are in the anonymous link. Due to cost and time limitations, large-scale human baselines are not yet feasible. Before that, we hope to ensure our framework is rigorous, effective, and credible.
> - Q1: It seems you're interested in the DC, which captures the impact of the Sheriff on others' decisions. The Ratio quantifies the credibility of the Sheriff. As you've pointed out, in our scenarios, most LLMs do not exhibit significant OL.
> - Q2:  DC cannot capture the situations you mentioned, but such scenarios are very rare. Our experiments show that in each round, less than 11\% of players (<1) agree with the Sheriff before it speaks, a percentage that may decrease with more players.
> - Q3: We propose a broader DC without requiring alignment with the Sheriff in the anonymous link, called DC*. Since general decision changes may be affected by other uncontrollable factors, our original DC captures a significant, yet partial, clean aspect of the Sheriff's OL.
> - Q4: Yes, the Sheriff can only change around 1 player's mind in each round. Most tested LLMs do not show strong OL, but our results indicate a trend that as LLMs advance, their OL improves.

---

> > ### Comment · Reviewer_fuFN · 2024-06-02
> > **Comment from reviewer**
> >
> > Thanks to the authors for the detailed response. They have clarified most of my concerns and are very helpful. Thanks!

---

> ### Author Response · Authors · 2024-06-06
> **Authors Response**
>
> Thank you for your prompt response. Your input is invaluable in enhancing our paper!

---

### Author Response · Authors · 2024-06-07
**Appreciation for Reviewers**

Dear Reviewers,

We would like to express our sincere gratitude for your prompt feedback and insightful suggestions. Your comments have played a crucial role in improving the clarity and quality of our paper.

In response to your suggestions, we will incorporate all additional analysis and experimental results during the rebuttal process into the final version of our paper. Furthermore, we will adjust the organization of our paper for greater readability. If you have any other questions or require further clarification, please do not hesitate to contact us.

Best regards,

Submission 406 Authors

---

### Decision · Program_Chairs · 2024-07-10

**Decision:**

Accept

**Comment:**

Four reviewers have provided insightful comments on the paper, and I have also read the paper carefully myself. The reviewers unanimously agree that this paper makes a valuable contribution by introducing a novel framework for studying opinion leadership in LLMs. The authors' approach is creative and well-motivated, and the proposed metrics and human evaluations strengthen the interpretability of the findings, which have practical implications for deploying LLMs in multi-agent interaction contexts.

The authors have provided thoughtful and detailed responses to the reviewers' comments and suggestions during the rebuttal period, addressing concerns about the impact of opinion leadership on game outcomes, the complexity of human social dynamics, and the overall organization of the paper. The authors have also provided further analyses and results which greatly clarify the contribution, including win rates, completion rates, and the performance of human players as the Sheriff.

While there are important improvements and clarifications to make based on the reviews, these concerns do not diminish the overall value of the work. The authors acknowledge these suggestions and plan to address them in the final version of the paper. I particularly encourage the authors to consider restructuring the paper to foreground information necessary to understand the key results (e.g. the WWQA dataset) and move some of the details from 3.1 ("LLM-based Player") to the appendix.

In summary, the novelty of the Werewolf framework, the thoroughness of the experiments, and the practical implications of the findings make this paper a strong accept.